# Ratiometric GPCR signaling enables directional sensing in yeast

**Nicholas T. Henderson**[1], **Michael Pablo**[2,3], **Debraj Ghose**[1], **Manuella R. Clark-Cotton**[1], **Trevin R. Zyla**[1], **James Nolen**[4], **Timothy C. Elston**[5], **Daniel J. Lew**[1]*

**1** Department of Pharmacology and Cancer Biology, Duke University, Durham, North Carolina, United States of America, **2** Department of Chemistry, University of North Carolina at Chapel Hill, Chapel Hill, North Carolina, United States of America, **3** Program in Molecular and Cellular Biophysics, University of North Carolina at Chapel Hill, Chapel Hill, North Carolina, United States of America, **4** Department of Mathematics, Duke University, Durham, North Carolina, United States of America, **5** Department of Pharmacology, University of North Carolina at Chapel Hill, Chapel Hill, North Carolina, United States of America

* daniel.lew@duke.edu

**Data Availability Statement:** Code and key data for Figs 5G, 6B, 8E, and S2–S4 Figs are available at https://github.com/mikepab/ratiometric-gpcr-particle-sims. Code for Figs 2G, 4D, 4E, 8B, and 8C is available at https://github.com/DebrajGhose/

## Abstract

Accurate detection of extracellular chemical gradients is essential for many cellular behaviors. Gradient sensing is challenging for small cells, which can experience little difference in ligand concentrations on the up-gradient and down-gradient sides of the cell. Nevertheless, the tiny cells of the yeast *Saccharomyces cerevisiae* reliably decode gradients of extracellular pheromones to find their mates. By imaging the behavior of polarity factors and pheromone receptors, we quantified the accuracy of initial polarization during mating encounters. We found that cells bias the orientation of initial polarity up-gradient, even though they have unevenly distributed receptors. Uneven receptor density means that the gradient of ligand-bound receptors does not accurately reflect the external pheromone gradient. Nevertheless, yeast cells appear to avoid being misled by responding to the fraction of occupied receptors rather than simply the concentration of ligand-bound receptors. Such ratiometric sensing also serves to amplify the gradient of active G protein. However, this process is quite error-prone, and initial errors are corrected during a subsequent indecisive phase in which polarity clusters exhibit erratic mobile behavior.

## Introduction

Chemical gradients provide cells with critical information about their surroundings, allowing them to navigate via chemotropism (gradient-directed growth) or chemotaxis (gradient-directed migration). For example, axons steer their growth up gradients of netrin to form new synapses, social amoebae crawl up gradients of cyclic adenosine monophosphate (cAMP) to aggregate into fruiting bodies, sperm swim up gradients of chemoattractants to find eggs, and neutrophils migrate up gradients of bacterial peptides or cytokines to eliminate pathogens from mammalian tissues [1–3]. In most cases, cells sense external signals via G-protein–coupled receptors (GPCRs), leading to cytoskeletal reorganization that produces directional growth or movement [4].

Ratiometric-GPCR-signaling-enables-directional-
sensing-in-yeast. All other relevant data are
available in S1 Data spreadsheet.

**Funding:** MRC-C is a Howard Hughes Medical
Institute Gilliam Fellow and holds a Graduate
Diversity Enrichment Program Award from the
Burroughs Wellcome Fund. This work was funded
by NIH/NIGMS grants R35GM127145 to TE and
R01GM103870 and R35GM122488 to DJL. The
funders had no role in study design, data collection
and analysis, decision to publish, or preparation of
the manuscript.

**Competing interests:** The authors have declared
that no competing interests exist.

**Abbreviations:** Bem1, bud emergence 1; Bni1, bud
neck involved 1; cAMP, cyclic adenosine
monophosphate; Cdc, cell division control; CDK,
cyclin-dependent kinase; CFP, cyan fluorescent
protein; CP, clustering parameter; CSM, complete
synthetic medium; CTM, carboxy-terminal
transmembrane domain; CV, coefficient of
variation; CXCR2, CXC chemokine receptor 2; Far1,
factor arrest 1; FRAP, Fluorescence Recovery After
Photobleaching; Fus3, fusion 3; GAP, GTPase-
activating protein; GEF, guanine nucleotide
exchange factor; GFP, green fluorescent protein;
GPAAD, GPFAD to GPAAD mutation; GPCR, G-
protein–coupled receptor; hsRGS4, *Homo sapiens*
regulator of G-protein signaling 4; Kss1, kinase
suppressor of supersensitive 2 mutations 1;
MAPK, mitogen-activated protein kinase; MAT,
mating type; MFα1, major α-factor gene 1; NLS,
nuclear localization sequence; NPF, GPFAD to
NPFAD mutation; $P_{gal1}$, galactose metabolism 1
promoter; RGS, regulator of G-protein signaling;
ROI, region of interest; sf, superfolder; Snc2,
suppressor of the null allele of CAP 2; Spa2,
spindle pole antigen 2; SR, Signal Ratio; Sst2,
supersensitive 2; Ste, sterile; tdTomato, tandem
dimer tomato; TEF1, translation elongation factor 1;
$T_{ic}$, time of initial clustering; $T_p$, time of polarization;
v-SNARE, vesicle-soluble NSF attachment protein
receptor; YEPD, 1% yeast extract, 2% peptone, 2%
dextrose; 7XR, 7 lysine to arginine mutations.

The sequence of molecular events that transduce extracellular signals to produce gradient-directed outputs is perhaps best understood in the genetically tractable budding yeast *S. cerevisiae*. Yeast are nonmotile unicellular fungi, and haploid yeast cells of mating type (MAT) **a** can mate with haploids of MATα. The cells secrete peptide pheromones that bind GPCRs on cells of the opposite mating type (α-factor is sensed by sterile 2 [Ste2] in **a** cells, and **a**-factor is sensed by Ste3 in α cells) [5]. Pheromone-bound receptors activate heterotrimeric G proteins to generate GTP–Gα and Gβγ. Gβγ recruits two key scaffold proteins, Ste5 and factor arrest 1 (Far1), from the cell interior to the membrane (Fig 1A) [6–8]. Ste5 recruitment leads to activation of the mitogen-activated protein kinases (MAPKs) fusion 3 (Fus3) and kinase suppressor of supersensitive 2 mutations 1 (Kss1), which induce transcription of mating-related genes, arrest the cell cycle in G1 phase in preparation for mating, and promote cytoskeletal polarization [8]. Far1 recruitment orients the cytoskeleton towards the mating partner by providing spatial information to the conserved Rho-family GTPase cell division control 42 (Cdc42), which is the master regulator of cell polarity in yeast (Fig 1A) [6,7,9,10].

To establish a polarized axis, Cdc42 becomes concentrated and activated at a site on the cell cortex referred to as a "polarity patch" [9]. Localized GTP–Cdc42 then acts through formins to orient linear actin cables towards the site, and the cables deliver secretory vesicles that mediate local growth and fusion with a mating partner [11–14]. Polarity establishment is thought to involve a positive feedback loop whereby local GTP–Cdc42 promotes activation of further Cdc42 in its vicinity [15]. Cdc42 is activated by the guanine nucleotide exchange factor (GEF) Cdc24 [16], which is recruited to the polarity patch by the scaffold protein bud emergence 1 (Bem1), which is itself recruited to the patch by Cdc42 effectors, providing a mechanism for positive feedback [17]. Cdc24 also binds directly to Far1, and the Gβγ–Far1–Cdc24 complex is thought to enhance GEF-mediated Cdc42 activation at sites with elevated levels of Gβγ [6,7,10,18]. Mutations that disrupt Far1–Cdc24 binding do not affect polarity establishment per se, but they abolish the ability to properly orient polarity with respect to the pheromone gradient [6,7]. Thus, Far1 provides a direct spatial connection between upstream receptor–pheromone binding and downstream Cdc42 activation, allowing the cells to exploit the pheromone gradient to find their partners.

Like other eukaryotic cells, yeast cells are thought to compare the ligand concentrations across the cell to determine the orientation of the gradient [19]. If the distribution of pheromone-activated receptors reflects the pheromone gradient, then Gβγ–Far1–Cdc24 complexes will be enriched up-gradient, spatially biasing activation of Cdc42 to kick off positive feedback at the right location for mating. However, the accuracy of such global spatial gradient sensing is limited by the small yeast cell size (approximately 4 μm diameter) [20], and simulations constrained by experimental data on binding and diffusion parameters suggested that the process would be inaccurate [21]. Indeed, when yeast cells are exposed to artificial, calibrated pheromone gradients, polarized growth often starts in the wrong direction [22,23]. Such cells can nevertheless correct initial errors by moving the polarity site [24,33,81].

Moving a Cdc42 patch that is constantly being reinforced by positive feedback seems counterintuitive, but time-lapse imaging revealed that the patch "wandered" around the cortex of pheromone-treated cells on a several-minute timescale [24]. Wandering was dependent on actin cables and vesicle traffic, which perturb the polarity patch [24–26]. New pheromone receptors are delivered to the polarity site, and after binding pheromone, the receptors are rapidly internalized and degraded [27–30]. As a result, pheromone receptors and their associated G proteins become concentrated in the vicinity of the polarity site, generating a sensitized region of membrane that can detect the local pheromone concentration [26,31,32]. As the polarity site wanders around the cortex, this receptive "nose" would sample pheromone levels at different locations.

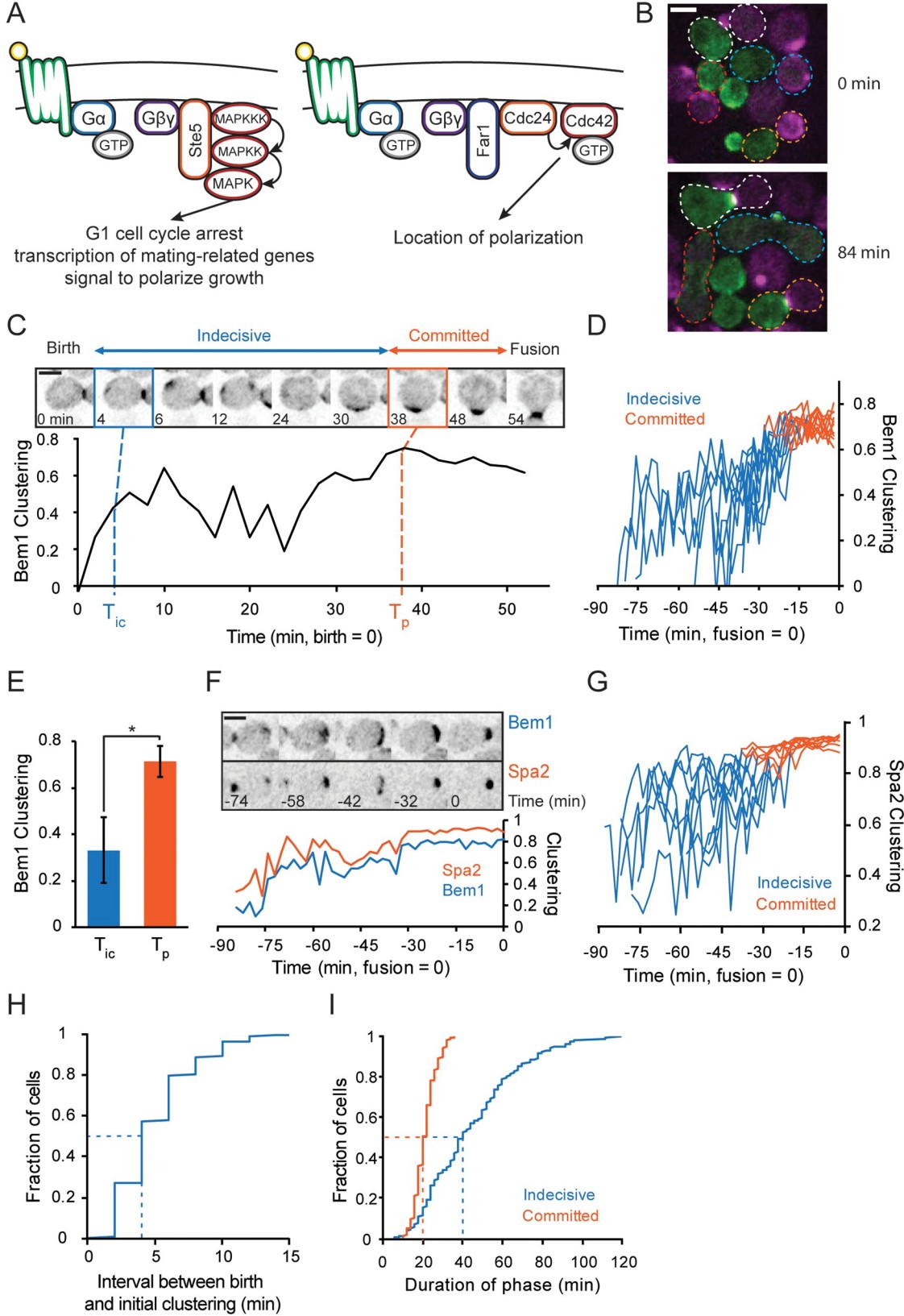

**Fig 1. Indecisive and committed phases of polarity behavior in mating yeast.** (**A**) Pheromone (α or **a**-factor, yellow) binds to pheromone receptors (Ste2 or Ste3, green), activating receptor-associated Gα and Gβγ. (Left) Gβγ recruits Ste5, which activates a MAPK cascade leading to cell cycle arrest in G1, transcription of mating-related genes, and polarization. (Right) Gβγ recruits Far1, which is bound to the GEF Cdc24, leading to local Cdc42 activation. (**B**) Representative images of cells with fluorescent polarity probes in a mating mixture (**a**, Bem1–tdTomato, magenta; α, Bem1–GFP, green). At 0 min (top), cells are budding; those that will go on to mate are circled. The same mating pairs are indicated at 84 min (bottom). By this time, two pairs have fused, forming zygotes with mixed magenta/green fluorescence (outlined in red, blue), and two pairs have polarized toward one another but not yet fused (outlined in white, orange). (**C**) Localization of Bem1–GFP in a representative mating cell. Top: Inverted maximum z-projection images of Bem1–GFP at selected times (cytokinesis = 0 min). A weak Bem1 cluster appears 4 min (blue box). The cluster moves and fluctuates in intensity during an "indecisive phase" until 38 min (orange box), when it strengthens and remains stationary during a "committed phase" until fusion occurs at 54 min. Bottom: quantification of Bem1 clustering (CP) in the same cell (see Materials and Methods). (**D**) Bem1 CP in 10 representative mating cells, as in (C). The timeline extends back to the time of cell birth from the time of fusion (0 min). Color switches from blue to orange at $T_p$. (**E**) Bem1 CP at $T_{ic}$ and $T_p$ ($n = 44$, error bars = SD, $^*t$ test, $p < 0.05$). (**F**) Localization of Bem1–GFP and Spa2–mCherry in a mating cell from birth (−82 min) to fusion (0 min). Top: Inverted maximum z-projection images of the indicated probes. Bottom: quantification of Bem1 and Spa2 CP in the same cell. (**G**) Spa2 CP in nine representative cells, displayed as in (D). (**H**) The cumulative distribution ($n = 246$) of the interval between birth and $T_{ic}$ in mating cells. (**I**) The cumulative distribution ($n = 246$) of the duration of the indecisive (blue) and committed phase (orange). Dashed lines indicate median. Scale bar, 3 μm. Strains: DLY12943, DLY7593 (B–E, H, I), DLY9070 (E), DLY21379 (F, E). Bem1, bud emergence 1; Cdc, cell division control; CP, clustering parameter; Far1, factor arrest 1; GEF, guanine nucleotide exchange factor; GFP, green fluorescent protein; MAPK, mitogen-activated protein kinase; Spa2, spindle pole antigen 2; Ste, sterile; tdTomato, tandem dimer tomato; $T_{ic}$, time of initial clustering; $T_p$, time of polarization.

If a cell's polarity site were facing a potential partner, then the cell would detect a higher pheromone concentration than if the polarity site were misoriented. In yeast cells engineered to activate high levels of MAPK activity by artificial recruitment of the scaffold Ste5 to the plasma membrane, the cells develop a strong polarity site that wanders gradually around the cell surface. When such cells are exposed to high levels of pheromone, the patch stops moving [26]. These observations suggested an "exploratory polarization" hypothesis to explain error correction: movement of the cell's polarity site would continue until the cell sensed a high pheromone level, indicating that the site was correctly oriented towards a mating partner [33].

A behavior strikingly similar to the exploratory polarization strategy discussed above was described for mating cells of the distantly related fission yeast *Schizosaccharomyces pombe* [34,35]. Unlike budding yeast, which mate rapidly in nutrient-rich conditions, fission yeast under starvation conditions. Potential mating partners exhibit an extended period in which they sequentially assemble and disassemble a weak polarity cluster at multiple locations. Clusters that happen to assemble in the vicinity of a cluster from a mating partner become strengthened and stabilized, presumably because of detection of a higher pheromone level. This strategy has been termed "speed dating,"

The extent to which yeast cells rely on global spatial sensing to orient the formation of a polarity site, versus exploratory polarization after the site has formed, remains unclear. A recent study found that when cells were placed in an artificial pheromone gradient in a micro-fluidics device, initial site formation was essentially random with respect to the gradient, and orientation occurred almost entirely by exploratory polarization [36]. However, it is unclear whether similar results might apply to different pheromone gradients or to more physiological conditions in which gradients are generated by mating partners.

To better understand how yeast actually locate their mating partners, we imaged mating events in mixed populations of **a** and α cells. We found evidence for both global spatial sensing and error correction. Encounters between partners were characterized by (i) rapid and non-random initial clustering of polarity proteins biased towards the partner; (ii) an "indecisive phase," in which fluctuating polarity sites relocalized in an erratic and very dynamic manner; and (iii) a "committed phase," in which cells polarized stably towards mating partners, culminating in fusion. Transition from indecisive to committed behavior was associated with a rise in MAPK activity. Initial polarization was surprisingly accurate given that it occurred despite a highly nonuniform (and thus potentially misleading) distribution of receptors. We found that

cells were able to compensate for the variation in receptor density via "ratiometric" sensing of the ratio of occupied versus unoccupied pheromone receptors across the cell [37]. Moreover, such ratiometric sensing improved the accuracy of gradient detection and provided a measure of gradient amplification. These findings reveal how yeast cells can overcome the challenges imposed by small cell size and lack of cell mobility to locate mating partners.

## Results

### Indecisive and committed phases of mating cell polarization

To observe how cells find their mates, we mixed **a** and α cells expressing differently colored polarity probes, Bem1-GFP (α) and Bem1-tandem dimer tomato (tdTomato) (**a**), and imaged them at 2 min resolution (S1 Video and Fig 1B). Fusion events were identified from movies and the cells were tracked back to their time of "birth" (the cytokinesis that preceded the mating event). Fig 1C (top) illustrates selected frames from a representative mating cell. This cell formed a faint initial cluster of Bem1 just 4 min after birth (blue panel), which then fluctuated in intensity and moved erratically around the cell cortex for 34 min before stably polarizing adjacent to a mating partner (orange panel). After another 16 min, the two cells fused, as seen by the mixing of red and green probes. We designate the time between initial cluster formation ($T_{ic}$) and stable polarization ($T_p$) as the "indecisive phase," reflecting the erratic behavior of the polarity probe. We designate the time between stable polarization ($T_p$) and fusion as the "committed phase" of mating, reflecting the strong and stably located polarity site. We note that cell shape remained approximately constant during the indecisive phase, with growth of a mating projection occurring only during the committed phase.

To quantify the degree of Bem1 polarization, we used a metric that exploits the pixel intensity distribution within the cell to assess the degree of signal clustering (clustering parameter, hereafter CP: see Materials and Methods). In mating cells, Bem1 CP was initially low at $T_{ic}$, fluctuated during the indecisive phase, increased near $T_p$, and then remained high throughout the committed phase (Fig 1C–1E). A similar two-stage process was observed for cells expressing fluorescent versions of spindle pole antigen 2 (Spa2), a polarisome component that binds and helps to localize the formin bud neck involved 1 (Bni1) (S2 Video) [38–40]. Analysis of cells expressing both Bem1 and Spa2 probes revealed that although Spa2 clusters were more tightly focused than Bem1 clusters, the probes clustered, dispersed, and moved together (Fig 1F and 1G). We conclude that cells undergo a reproducible pattern of polarization during mating, with sequential indecisive and committed phases.

The earliest observable clustering of polarity factors occurred shortly after birth (Fig 1H: median 4 min after initiating cytokinesis), frequently at a different location than that of the final stable polarization (see below). The duration of the indecisive phase (Fig 1I: median 42 min) was very variable, ranging from 10 to 120 min. This is consistent with a search process that would take a variable amount of time depending on the availability and proximity of potential mating partners. In contrast, the committed phase was consistently about 20 min (Fig 1I), perhaps reflecting the time required to remodel the local cell walls and fuse.

### Timing of commitment

In our protocol, cells of each mating type are proliferating asynchronously before they are abruptly mixed. Thus, cells often first "see" each other while they are budding, and one cell of each mating pair enters G1 phase (where mating is possible) before the other. Because partner fusion during mating occurs at the same time for both partners, the first-born partner must extend one or both phases of polarization while the second-born partner completes the previous cell cycle and catches up (Fig 2A). Does the first-born locate and commit to its partner

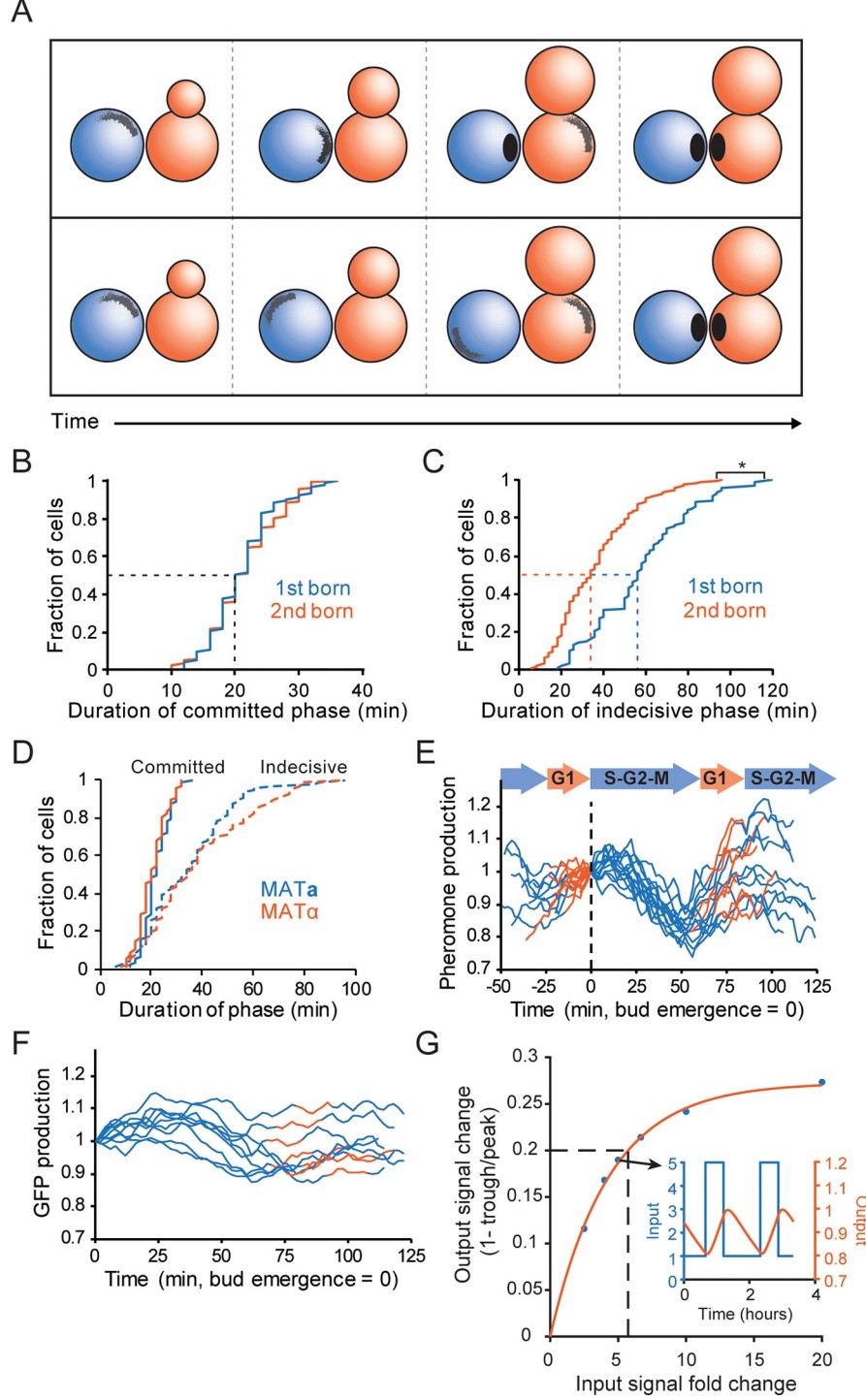

**Fig 2. Timing of commitment by mating partners.** (**A**) Two possibilities for polarization timing in mating partners that "meet" when they are at different stages of the cell cycle. Top: The first-born cell (blue) locates the partner (orange) while the latter is still completing the cell cycle. The first-born cell polarizes and waits during an extended committed phase for its partner to catch up. Bottom: The first-born cell cannot locate its partner until the partner enters G1 phase. It remains in an extended indecisive phase until the partner enters G1, after which both cells polarize. Dashed line: median. (**B**) Cumulative distribution of the interval between stable polarization and fusion (committed phase) in first-born (blue, $n = 93$) and second-born cells (orange, $n = 153$). (**C**) Cumulative distribution of the interval between initial clustering and commitment (indecisive phase) in first-born (blue, $n = 93$) and second-born cells (orange, $n = 153$) (*two-sample KS test, $p < 0.05$). Dashed lines: median. (**D**) Cumulative distribution of the duration of the indecisive

phase (dashed) and committed phase (solid) in second-born cells of MAT**a** (blue, $n = 87$) and α (orange, $n = 66$). (**E**) Pheromone synthesis is high in G1 and decreases in S/G2/M. Cells (MATα) expressing sfGFP from the MFα1 promoter were imaged for 150 min. Reporter fluorescence was normalized to the value at the time of first bud emergence (0 min: black dashed line). Curves were colored orange from birth to bud emergence (G1 phase) and blue from bud emergence to cytokinesis (S, G2, and M phases). (**F**) A reporter driven by the constitutive TEF1 promoter was analyzed as in E. (**G**) Modeling MFα1 promoter activity as switching between high in G1 and low in S/G2/M (inset, blue) can predict the expected fluctuation in fluorescence intensity (inset, orange). Main graph: fractional change in output fluorescence intensity signal ($y$ axis) predicted as a function of fold-change in the input MFα1 promoter activity across the cell cycle ($x$ axis). The observed 20% change (0.2: black dashed line) would require >5-fold change in pheromone synthesis rate (inset). Code for Fig 2G is available at https://github.com/DebrajGhose/Ratiometric-GPCR-signaling-enables-directional-sensing-in-yeast. Strains: DLY12943, DLY7593 (B-D), DLY22883 (E), DLY22928 (F). GFP, green fluorescent protein; KS, Kolmogorov–Smirnov; MAT, mating type; MFα1, major α-factor gene 1; sf, superfolder; TEF1, translation elongation factor 1.

first, and then wait (Fig 2A, top), or does the first-born remain indecisive until the second-born has caught up (Fig 2A, bottom)? We found no difference in the average duration of the committed phase between first and second-born cells (Fig 2B). Conversely, the indecisive phase was significantly longer in first-born cells (Fig 2C), suggesting that first-born cells remain indecisive while second-born cells complete the cell cycle and that cells only polarize stably towards partners that are in G1 (Fig 2A, bottom). We found no difference in the duration of either phase between **a** and α cells (Fig 2D).

The simplest hypothesis to explain why commitment is delayed until both cells are in G1 would be that pheromone secretion changes when cells enter G1. To assess the rate of pheromone synthesis, we introduced a fluorescent reporter whose production was driven by the promoter of the major α-factor gene (MFα1) [41,42]. Reporter signal varied regularly through the cell cycle, rising in G1 and falling (because of dilution) after bud emergence (Fig 2E). A control reporter expressed from the constitutive translation elongation factor (*TEF1*) promoter did not display a similar sawtooth fluctuation through the cell cycle (Fig 2F). The magnitude of the cell cycle variation in MFα1 reporter intensity was small (about 20%), but one would expect a stable reporter to integrate (and hence dampen) the degree of real variation. Modeling the expected effect of GFP fluorescence maturation and dilution (see Materials and Methods) suggested that the true α-factor synthesis rate (and presumably the rate of α-factor secretion) was over 5-fold higher during G1 than in S/G2/M (Fig 2G). Thus, first-born cells would detect lower levels of pheromone until their partners entered G1, and stable polarization towards a partner (commitment) may be triggered by increased pheromone signaling. This conclusion is consistent with earlier findings that the α-factor dose required to induce projection formation was significantly higher than that required to induce cell cycle arrest in G1 [43].

## Relation between MAPK activity and polarity factor dynamics

One consequence of pheromone signaling is the activation of the mating MAPKs Fus3 and Kss1 [5]. MAPK activity is necessary for projection formation [44], and MAPK activity may also be sufficient to induce projection formation because artificial membrane recruitment of Ste5 (promoting high MAPK activity) leads to projection formation even in the absence of pheromone [8,45]. Moreover, MAPK activity in mating cells increases as cells begin to form projections [44]. Given these findings, it seemed likely that an increase in MAPK activity might be responsible for triggering the change from indecisive to committed behavior of the polarity site. To monitor MAPK activity in mating cells, we introduced a single-cell MAPK sensor [46] into our strains together with the Spa2 probe. The MAPK sensor is a fluorescent probe that moves from the nucleus to the cytoplasm when it is phosphorylated by active MAPK. In the absence of pheromone, the sensor was predominantly nuclear, although the

nuclear-to-cytoplasmic ratio varied somewhat through the cell cycle, peaking during anaphase (Fig 3A and S3 Video). We suspect that the sensor may be phosphorylated by the cyclin-dependent kinase (CDK) during the cell cycle because nuclear export of the sensor peaks in mitosis, and MAPK and CDK are both proline-directed kinases that often overlap in substrate specificity. In a mating mix, the sensor distribution became uniform prior to fusion, reflecting the expected increase in MAPK activity (Fig 3B and S4 Video). To quantify the degree of nuclear concentration of the MAPK sensor, we measured the coefficient of variation (CV) in pixel intensity across the cell. When the probe is nuclear, the bright nuclear and dim cytoplasmic pixels yield a high CV, but when the probe distribution is uniform, there is a low CV. We found considerable cell-to-cell variability in this signal (S1A Fig), which could be largely accounted for by differences in the level of expression of the probe (S1B Fig). We developed a MAPK activity metric based on the CV of the probe (S1C and S1D Fig).

In mating cells, MAPK activity fluctuated but then climbed to a plateau about 20 min prior to fusion (Fig 3C). Because this was similar to the clustering behavior of polarity probes, we directly compared MAPK activity with Spa2 CP in individual mating cells (Fig 3D). These measures aligned well with one another in most cells, with both Spa2 CP and MAPK activity fluctuating during the indecisive phase before rising to a stable plateau during the committed phase (Fig 3D and 3E). However, because the apparent plateau may be influenced by saturation of the probe's dynamic range, our data do not exclude the possibility that MAPK activity continues to rise further during the committed phase. Indeed, a study using a different FRET-based probe [44] suggested that MAPK activity reached a peak just before fusion.

To more directly ask whether an increase in MAPK activity promotes stable polarization, we induced MAPK activity in the absence of pheromone by expressing a membrane-tethered version of the MAPK scaffold Ste5 [8]. As MAPK activity increased and the MAPK sensor exited the nucleus, Spa2 switched from faint and mobile clustering to become strongly polarized (Fig 3G). The timing varied from cell to cell, but nearly all cells (97%, $n = 118$) with induced MAPK eventually formed strong polarity patches. Thus, intermediate MAPK activity is associated with indecisive polarity factor dynamics, whereas elevated MAPK signaling is associated with strong polarization.

A cross-correlation analysis of Spa2 CP and MAPK activity during the indecisive phase revealed that they fluctuated in tandem (Fig 3F). This correlation suggests that varying MAPK activity might contribute to fluctuations in polarity factor clustering, that polarity factor clustering might influence MAPK activity, or both (see Discussion).

## Gradient sensing before initial polarity clustering

Our findings thus far suggest that mating cells undergo the following sequence of events. Newborn cells in G1 increase their rate of pheromone production, signaling to potential partners that they are ready to mate. They arrest the cell cycle and initiate a weak level of polarization. There follows an indecisive phase, during which polarity factors cluster at mobile and unstable locations. At some point, MAPK activity increases, and polarity factors become stably concentrated and oriented towards a partner.

The indecisive phase may allow exploratory polarization to locate partners, as suggested by studies with cells exposed to uniform pheromone [24,26] or artificial gradients [36]. Exploratory polarization differs markedly from the traditional spatial sensing model, in which unpolarized G1 cells first sense the pheromone gradient and only then polarize, generally in the right direction [19,31,47,48]. These views are not mutually exclusive, and it could be that significant gradient sensing takes place prior to the initial clustering of polarity factors. Indeed, we found that in our mating mixtures, cells biased the locations of their initial polarity clusters

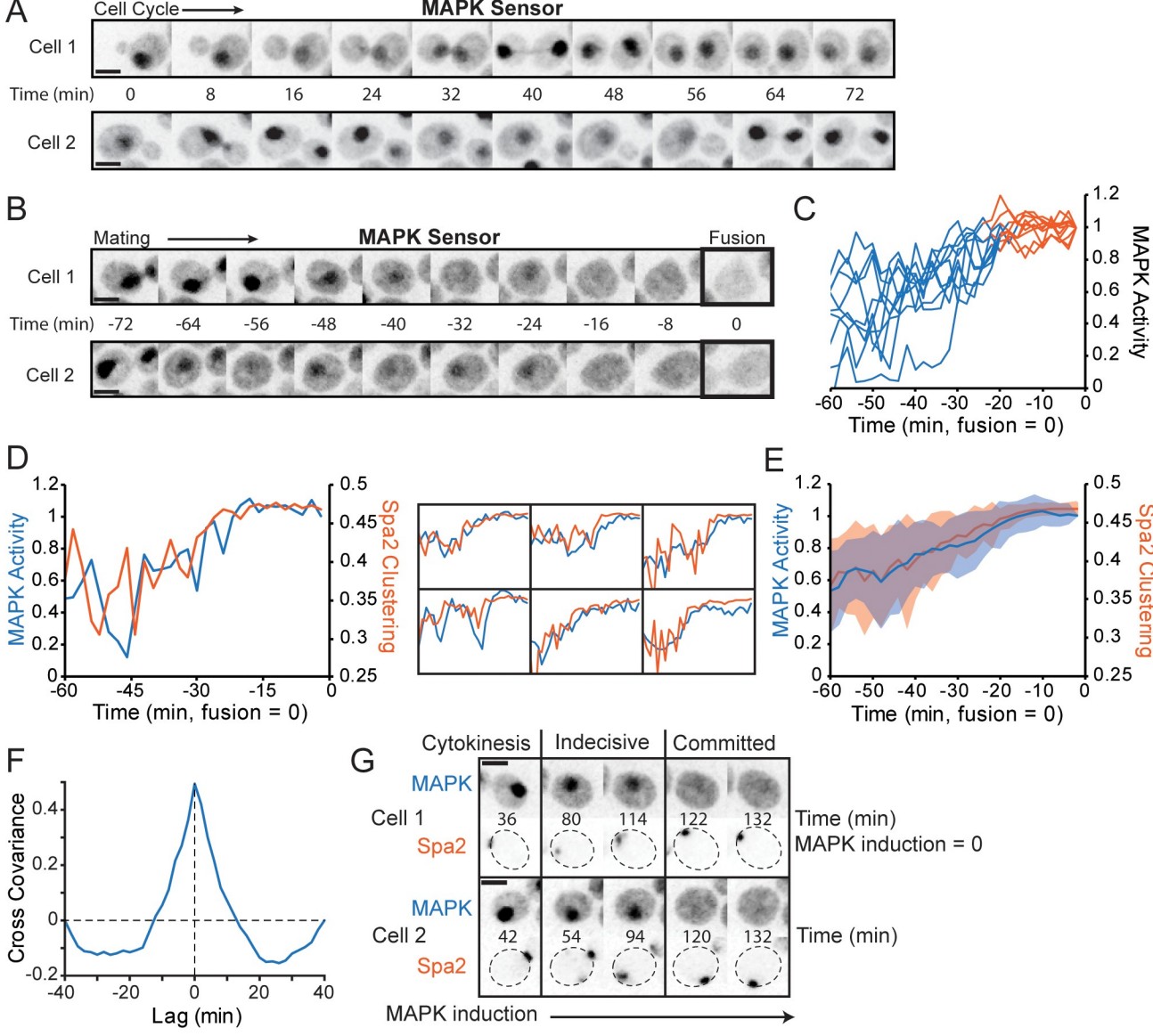

**Fig 3. Commitment coincides with an increase in MAPK activity.** (**A**) Localization of MAPK activity sensor varies through the cell cycle. Inverted maximum z-projection images of the sensor Ste7$_{1-33}$–NLS–NLS–mCherry in representative vegetatively growing cells. (**B**) The sensor is exported from the nucleus in response to MAPK activation. MAT**a** cells harboring Ste7$_{1-33}$–NLS–NLS–mCherry were mixed with MATα cells and imaged as in (A). Representative mating cells are illustrated from birth to fusion (0 min = fusion). (**C**) MAPK activity calculated from sensor distribution (see Materials and Methods) in the 60 min prior to fusion for 10 representative cells. The transition from the indecisive (blue) to committed phase (orange) was determined from a Spa2-GFP probe in the same cells. (**D**) Left: MAPK activity (blue, as in C) and Spa2 CP (orange, as in Fig 1F) in a representative mating cell. Right: six other cells. (**E**) Average MAPK activity (blue) and Spa2 CP (orange) in the 60 min prior to fusion (*n* = 41 cells). Shading: SD. (**F**) Cross-covariance of MAPK activity and Spa2 CP during the indecisive phase (window from 60 min to 20 min before fusion) in mating cells (*n* = 41 cells). Lag represents the time by which the Spa2 CP was shifted forward in time relative to the MAPK activity. 1 = perfect cross-covariance. (**G**) Cells harboring P$_{gal1}$–Ste5–CTM allow MAPK induction by β-estradiol without pheromone treatment. Cells with the MAPK sensor and Spa2-GFP were imaged following β-estradiol treatment. Inverted maximum z-projection images of selected time points show Spa2 neck localization during cytokinesis, indecisive behavior upon intermediate MAPK activation, and committed behavior following high MAPK activation in representative cells. Scale bar, 3 μm. Strains: DLY22259 (A-F), DLY22764 (G). CP, clustering parameter; CTM, carboxy-terminal transmembrane domain; GFP, green fluorescent protein; MAPK, mitogen-activated protein kinase; MATα, mating type α; NLS, nuclear localization sequence; P$_{gal1}$, galactose metabolism 1 promoter; Spa2, spindle pole antigen 2; Ste, sterile.

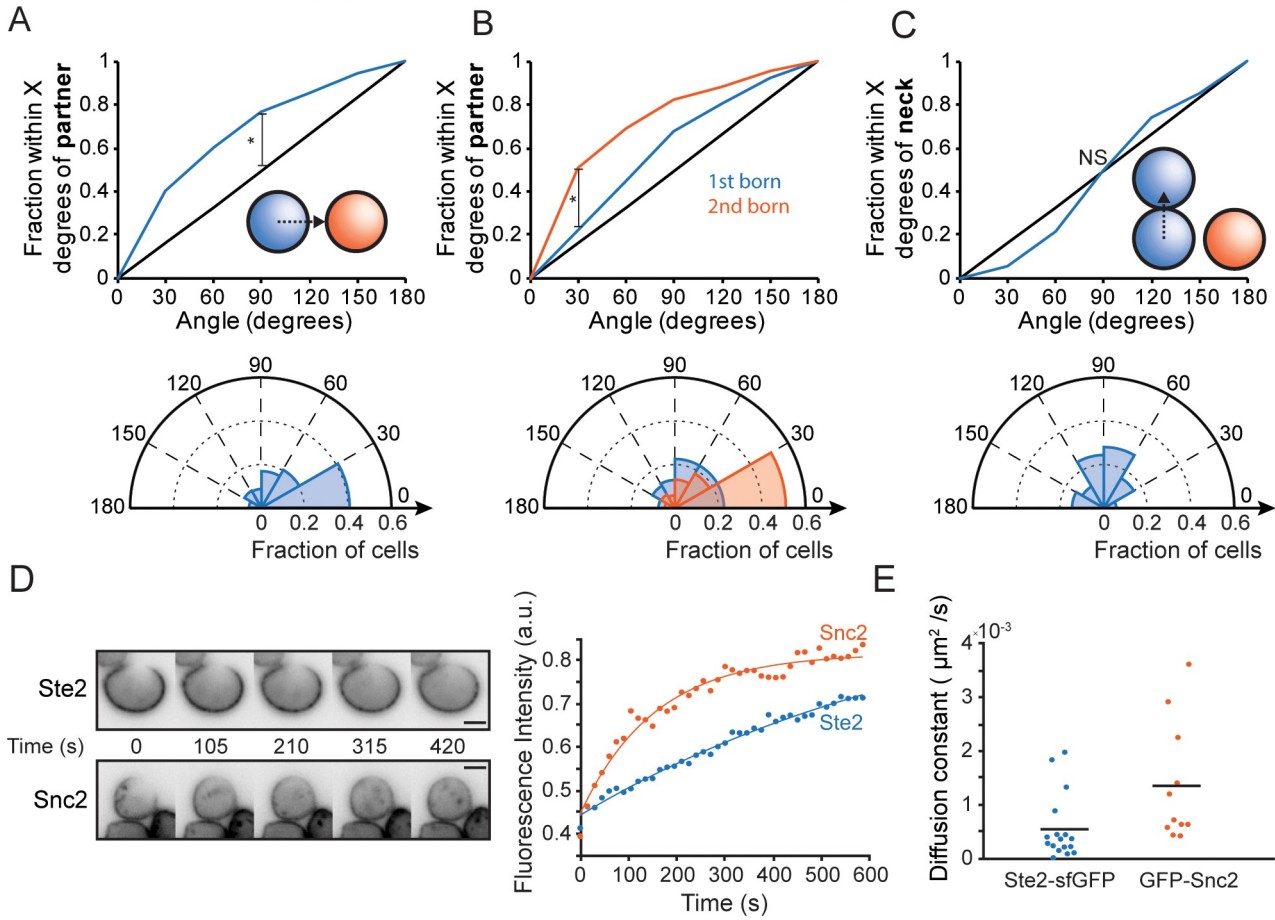

**Fig 4. Nonrandom initial clustering of polarity factors.** (**A**) Orientation with respect to partner. Top: cumulative distribution of initial Bem1 cluster location (inset: 0° = cluster oriented towards mating partner) relative to the mating partner (*n* = 246). Black line: hypothetical random distribution (*KS test, *p* < 0.05). Bottom: polar histogram display of the same data. (**B**) Top: cumulative distribution of initial cluster location relative to the mating partner, plotted separately for first-born (blue, *n* = 93) and second-born (orange, *n* = 153) cells (*KS test, *p* < 0.05). Bottom: polar histogram display of the same data. (**C**) Orientation with respect to neck. Cumulative distribution of initial cluster location relative to the site of cytokinesis plotted as in (A) (*n* = 246, KS test, NS). Bottom: polar histogram display of the same data. (**D**) (Left) Single-plane inverted fluorescent images of representative Ste2[7XR-GPAAD]–sfGFP (top) and Snc2-GFP (bottom) cells that were bleached to assess fluorescence recovery due to diffusion. (Right) Fluorescence recovery of the bleached region in the depicted cells, with exponential fits. (**E**) Estimated diffusion constants; each dot is one cell, and horizontal lines mark averages (*n* = 17 and 11 for Ste2–sfGFP and Snc2–GFP cells, respectively). Code for Fig 4D and 4E is available at https://github.com/DebrajGhose/Ratiometric-GPCR-signaling-enables-directional-sensing-in-yeast. Strains: DLY12943, DLY7593 (A–C), DLY21705, DLY17966 (D, E). Bem1, bud emergence 1; GFP, green fluorescent protein; GPAAD, GPFAD to GPAAD mutation; KS, Kolmogorov–Smirnov; NS, not significant; sf, superfolder; Snc2, suppressor of the null allele of CAP 2; Ste, sterile; 7XR, 7 lysine-to-arginine mutations.

towards their eventual mating partners (Fig 4A). This directional bias was significantly higher in second-born than first-born cells (Fig 4B), presumably because second-born cells are orienting towards partners that are already in G1 and secreting more pheromone. In contrast, we found no bias towards the previous cytokinesis site (neck) (Fig 4C). These findings suggest that spatial gradient sensing occurs prior to initial clustering.

How would spatial gradient sensing occur? We assume that the gradient signal would be interpreted via the spatial distribution of pheromone receptor complexes at the cell surface. However, diffusion of receptor–ligand complexes could degrade the information available to the cell regarding the spatial distribution of binding events. Simulations indicate that with diffusion similar to that estimated for single-pass transmembrane proteins in yeast (0.0025 μm²/s

[49]), such blurring would be problematic in extracting spatial information from a pheromone gradient [21]. The actual diffusion constant for yeast pheromone receptors is unknown.

We used Fluorescence Recovery After Photobleaching (FRAP) to estimate the Ste2 diffusion constant (see Materials and Methods). A representative FRAP experiment is shown in Fig 4D. The average estimated diffusion constant was 0.0005 μm$^2$/s (Fig 4E), about 5-fold slower than estimates for single-pass transmembrane proteins [49]. Control experiments with a vesicle-soluble NSF attachment protein receptor (v-SNARE) yielded more rapid diffusion (Fig 4D and 4E). Thus, yeast pheromone receptors diffuse very slowly, potentially allowing more accurate gradient sensing.

## Nonuniform pheromone receptor distribution

While slow receptor diffusion preserves spatial information regarding the location of pheromone binding events, slow diffusion also preserves receptor heterogeneity at the cell surface. In growing cells that were not exposed to α-factor, Ste2–superfolder (sf)GFP distribution varied through the cell cycle, accumulating around the neck during cytokinesis (Fig 5A and 5B). Ste2 distribution in G1 cells ranged from nearly uniform to highly polarized (Fig 5C). We assume that uneven receptors in newborn cells arise from polarized secretion towards the neck during cytokinesis. Quantification of surface Ste2 revealed a 3-fold difference (on average) in Ste2 concentration from one side of the cell to the other (Fig 5D).

The nonuniform receptor distribution poses a potential problem for accurate gradient sensing: one might expect cells to be preferentially sensitive to pheromone on the side where receptors are enriched, which might not correspond to the side facing a mating partner. To illustrate the problem, we conducted particle-based simulations of a model spherical cell with receptors distributed unevenly over the surface with a 3-fold difference from one side of the cell to the other (Fig 5E). We assume that heterotrimeric G proteins diffuse at the membrane and become activated when they encounter a ligand-bound receptor. G-protein inactivation occurs at a rate that is the same everywhere (see Materials and Methods). A stable pheromone gradient was simulated by assuming that the probability that a receptor is active is 1.5-fold higher on the up-gradient side than the down-gradient side of the cell (Fig 5F). Receptor diffusion would not significantly blur this gradient (S2 Fig). We recognize that pheromone gradients may not be stable under mating conditions, but in our model, we use them to allow for straightforward interpretation.

At any given time, our simulations provide the locations of all active G proteins on the cell surface. From that, we calculated a resultant vector for active G protein and plotted the angle between this vector and the imposed pheromone gradient (Fig 5G). When the receptor density gradient and the pheromone gradient were aligned, the simulated cells identified the correct direction (Fig 5G). However, when the density gradient was not aligned with the pheromone gradient, the simulated cells were easily misled, with active G proteins accumulating in regions with high receptor density (Fig 5G). Thus, without some compensatory mechanism, we would expect yeast cells to have difficulty decoding the pheromone gradient in the presence of an uncorrelated receptor density gradient.

## Ratiometric sensing of receptor occupancy

A potential compensatory mechanism that could correct for the presence of more active receptors at sites of high receptor density exploits the fact that receptor-dense regions would contain more unbound receptor as well as ligand-bound receptor. If unbound receptor counteracts G-protein activation, that could cancel out the higher rate of G-protein activation by ligand-bound receptors. In yeast cells, the regulator of G-protein signaling (RGS) protein

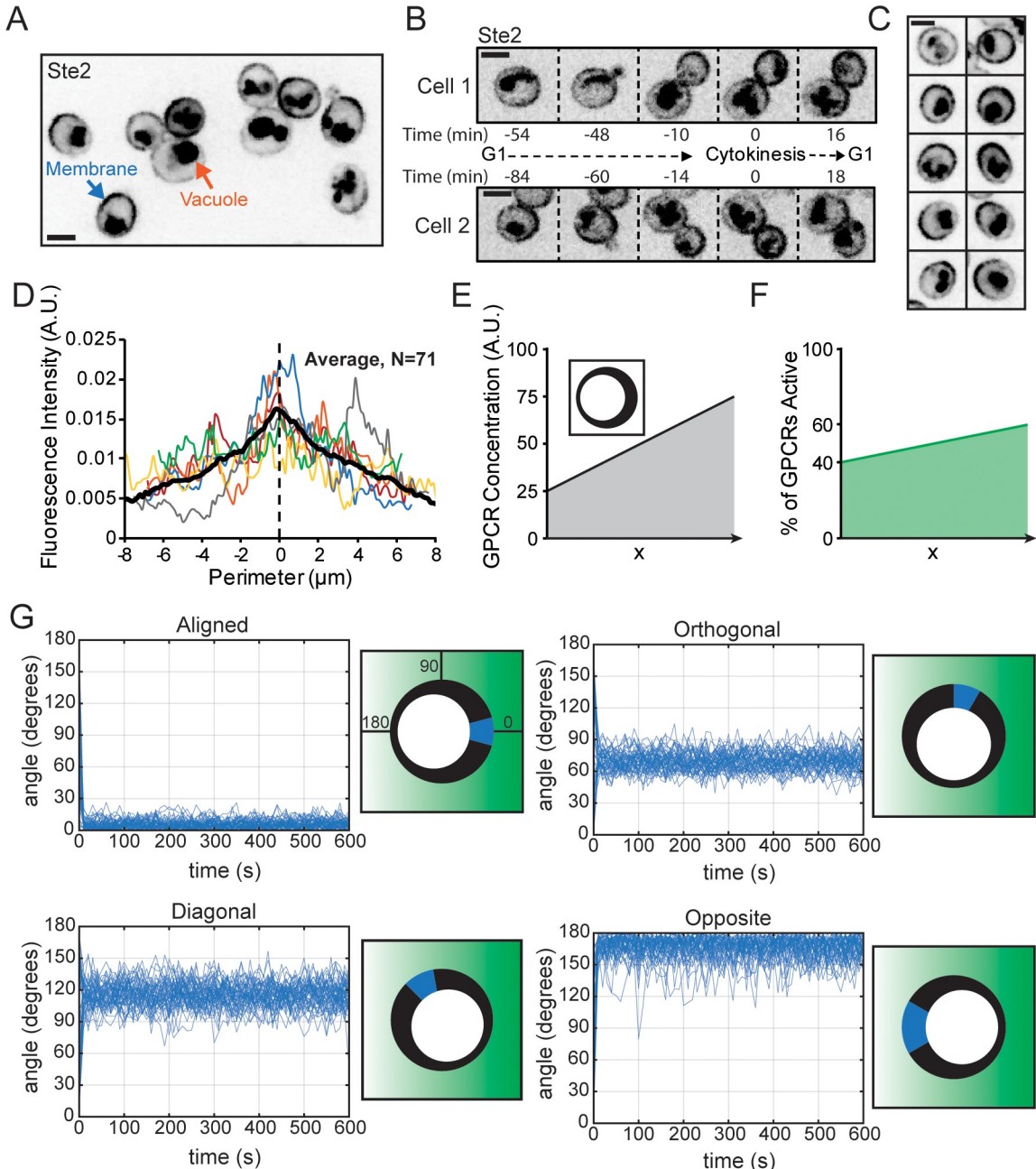

**Fig 5. Pheromone receptor density variation around the cell membrane.** (**A**) Single-plane inverted image of vegetatively growing cells expressing Ste2–sfGFP. Membrane signal (blue arrow) is Ste2–sfGFP, but vacuole signal (orange arrow) is probably sfGFP cleaved from Ste2-sfGFP after internalization. (**B**) Ste2 distribution through the cell cycle in representative cells. (**C**) G1 cells display Ste2 distributions ranging from almost uniform (top) to very asymmetric (bottom). (**D**) Quantification of Ste2–sfGFP membrane distribution in G1 cells. Individual linescans (examples in color) were normalized to have the same total fluorescence and centered on the peak of a smoothed spline fit. Black line, average ($n = 71$). (**E–G**) Particle-based simulations of receptor–G-protein interactions at the cell membrane. Receptors and G proteins were simulated as diffusing particles on a spherical surface. G proteins were activated when they encountered an active receptor, and active G proteins were spontaneously inactivated with first-order kinetics. (**E**) Receptors were distributed unevenly: receptor density is indicated by the thickness of the black line (inset) and reflects a 3-fold gradient, similar to the Ste2 distribution. (**F**) A 1.5-fold pheromone gradient was simulated along the $x$ axis by varying the % of active receptors from 40% to 60% across the cell diameter. (**G**) Simulations were conducted with receptor activity and density gradients oriented as in the illustrations. The locations of all of the active G proteins were used to calculate a G-protein vector, whose angle to the direction of the pheromone gradient is plotted ($y$ axis) against time ($x$ axis) (left). 0˚ indicates perfect orientation: active G-protein vector in the same direction as the applied receptor activity gradient. The approximate range of G-protein vectors (blue wedge) is shown in the cartoon on the right, along with the pheromone gradient (green shading) and receptor density (as in E). Code and key data for Fig 5G are available

at https://github.com/mikepab/ratiometric-gpcr-particle-sims. Scale bar, 3 μm. Strains: DLY20713 (A–D). A.U., arbitrary unit; GFP, green fluorescent protein; GPCR, G-protein–coupled receptor; sf, superfolder; Ste, sterile.

supersensitive 2 (Sst2) that inactivates the G protein is recruited to the cell membrane via binding to unoccupied Ste2 [50]. A recent study [37] insightfully suggested that this would cause cells to measure the ratio of ligand-bound to unbound receptors (i.e., ratiometric sensing). Pheromone-bound Ste2 loads GTP on Gα, whereas unbound Ste2–Sst2 promotes GTP hydrolysis by Gα, so the level of activated Gα depends on the ratio between pheromone-bound and unbound Ste2 rather than the absolute level of bound Ste2 (Fig 6A). Here, we explore the possibility that such ratiometric sensing would also lead to measurement of the spatial distribution of the ratio of active/total receptors so that differences in the local receptor density would not distort a cell's ability to determine the orientation of a pheromone gradient.

To test that idea, we repeated the simulations described above in which G proteins decode a pheromone gradient that is distorted by the presence of uneven receptor density. The central difference was that instead of a single G-protein deactivation rate regardless of spatial position, G-protein deactivation occurred when an active G protein encountered an unbound receptor. For fair comparison, the G-protein inactivation rate constants for the nonratiometric and ratiometric simulations were empirically calibrated using simulations to produce similar levels of active G protein at the midpoint of the gradient (Materials and Methods and S3A Fig). For the ratiometric model, the simulated cell correctly identified the direction of the pheromone gradient no matter what the receptor density distribution (Fig 6B). These simulations assumed that bimolecular reactions were diffusion-limited, but similar results were obtained assuming reaction-limited kinetics (S3B Fig). Thus, ratiometric sensing provides a robust mechanism for preventing cells from being misled by uneven receptor density.

To experimentally test the Sst2-based ratiometric sensing model, we replaced Sst2 with a human paralog, *Homo sapiens* regulator of G-protein signaling 4 (hsRGS4), which has similar GTPase-activating protein (GAP) activity towards Gα but does not associate with Ste2 [37]. hsRGS4 is myristoylated and localized uniformly to the plasma membrane (Fig 6C). We found that two copies of hsRGS4 expressed from the *SST2* promoter were sufficient to restore wild-type global pheromone sensitivity to cells lacking endogenous Sst2 (Fig 6D). However, cells with hsRGS4×2 were significantly worse at orienting at their initial Bem1 clusters towards their partners (Fig 6E). Instead, the initial clusters in hsRGS4x2 cells were strongly biased towards the previous mother-bud neck (Fig 6F), a region of high receptor density (Fig 5). Indeed, whereas wild-type cells often polarized Bem1 towards regions of low receptor density, hsRGS4×2 cells displayed a strong tendency to establish initial clusters of Bem1 at sites enriched for Ste2 (Fig 6G and 6H). Thus, gradient sensing depends on the endogenous RGS protein Sst2, which may assist in this process by linking Gα–GTP hydrolysis to the location of unbound receptor.

### Effect of changing receptor distribution

If the inaccurate gradient sensing exhibited by hsRGS4×2 cells is due to the uneven receptor density, then the orientation defect of hsRGS4×2 should be exacerbated in cells with more uneven receptors and corrected in cells with more uniform receptors (ratiometric sensing should be unnecessary if receptor density is uniform). To test this prediction, we generated cells that had different receptor distributions. To manipulate Ste2 distribution, we used Ste2 mutants that either lacked endocytosis signals (Ste2$^{7XR-GPAAD}$, where 7 lysine ubiquitination sites are mutated to arginine and the GPFAD endocytosis motif is mutated to GPAAD,

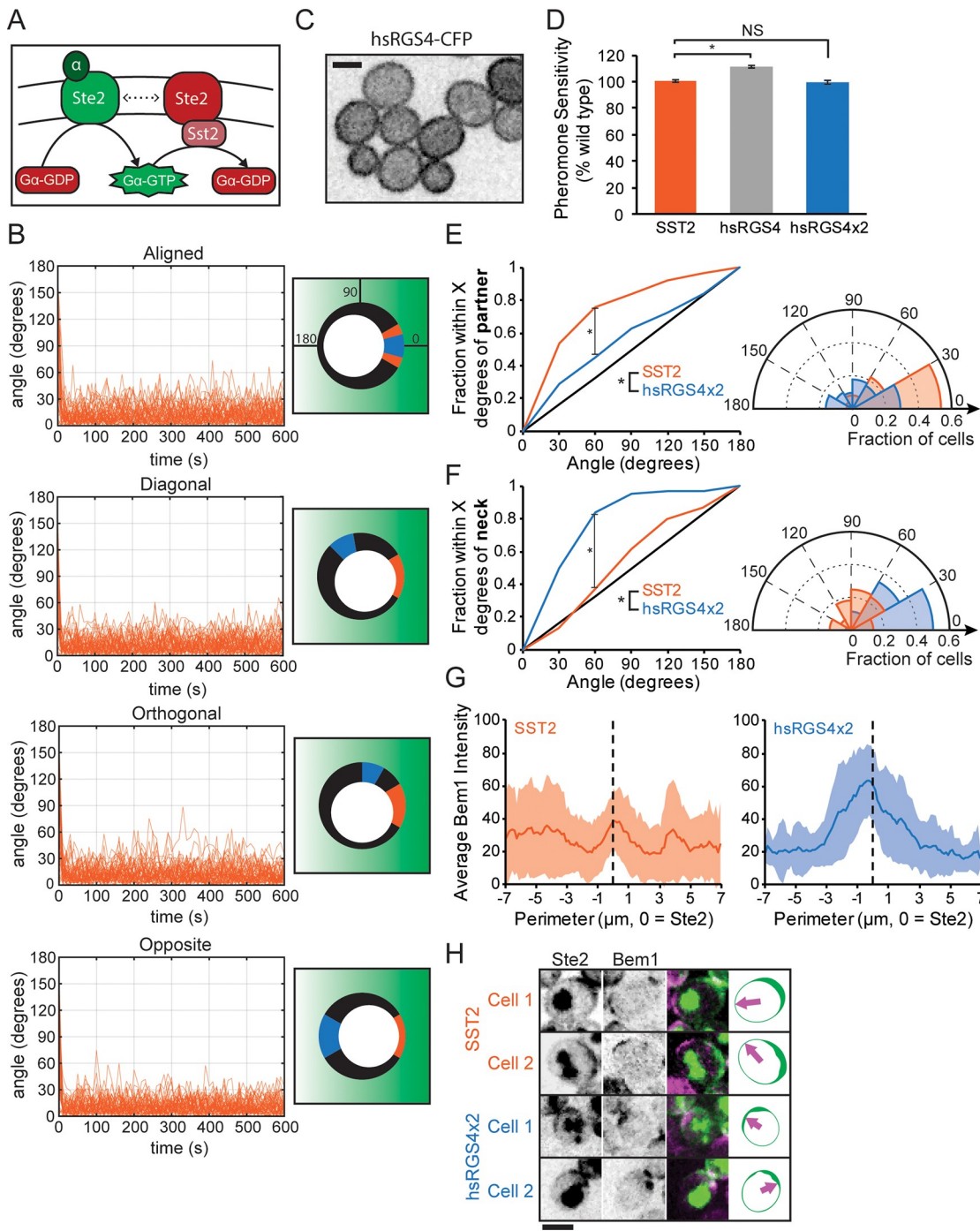

**Fig 6. Ratiometric sensing allows cells to orient towards partners despite uneven receptor density.** (**A**) Proposed ratiometric pheromone sensing mechanism. The G protein is activated by pheromone-bound receptor (Ste2 + α-factor) and inactivated by the RGS protein Sst2. Sst2 associates with inactive Ste2. Thus, G-protein activity reflects the ratio of bound to unbound receptors. (**B**) Particle-based simulations were repeated as in Fig 5, except that instead of spontaneous inactivation, G proteins were inactivated upon encountering inactive receptors. These "ratiometric" simulations (orange) were plotted as in Fig 5G. For comparison, both the nonratiometric (blue) and ratiometric (orange) results are depicted in the cartoons. (**C**) hsRGS4 is distributed uniformly on the membrane. Single-plane inverted image of hsRGS4–CFP. (**D**) Pheromone sensitivity measured via halo assay in wild-type cells (orange) and cells in which Sst2 was replaced by one copy (gray, hsRGS4, *t test, $p < 0.05$) or two copies (blue, hsRGS4×2, NS) of hsRGS4 ($n = 9$, three technical replicates at three pheromone concentrations, normalized to the average wild-type halo diameter). (**E**) Left: cumulative distribution of initial cluster location relative to the nearest potential mating partner in wild-type (orange, $n = 222$) and hsRGS4×2 cells (blue, $n = 62$, *two-sample KS test, $p < 0.05$). Right: polar histogram of the same data. (**F**) Left:

cumulative distribution of initial cluster location relative to the site of cytokinesis in wild-type (orange, $n = 222$) and hsRGS4x2 cells (blue, $n = 62$, *two-sample KS test, $p < 0.05$). Right: polar histogram of the same data. (**G**) Bem1 initial cluster location is biased by Ste2 distribution in cells with hsRGS4×2 but not in cells with wild-type Sst2. Averaged Bem1–tdTomato distribution (shaded region = standard deviation) at the time of initial clustering, centered on the location with maximum Ste2, in wild-type (orange, $n = 33$) and hsRGS4×2 cells (blue, $n = 33$). Ste2 and Bem1 linescans were acquired from maximum projection images. (**H**) Example images of cells at the time of initial Bem1 clustering. Bem1 clusters (magenta) sometimes form in areas depleted of receptor (green) in wild-type cells, but in hsRGS4×2 cells, clusters tend to form where receptors are concentrated. Single-plane Ste2–sfGFP images (first column), maximum projection Bem1–tdTomato images (second column), and overlays (third column) from representative wild-type and hsRGS4×2 cells. A simplified cartoon (fourth column) depicts Ste2 distribution and the location of Bem1 initial clusters (arrow). Code and key data for Fig 6B are available at https://github.com/mikepab/ratiometric-gpcr-particle-sims. Scale bar, 3 μm. Strains: DLY22318 (C, D), DLY22321 (D), DLY22520 (D–F), 12943 (E–F), DLY22243, 22628 (G, H). Bem1, bud emergence 1; CFP, cyan fluorescent protein; GFP, green fluorescent protein; hsRGS4, *Homo sapiens* regulator of G-protein signaling 4; KS, Kolmogorov–Smirnov; NS, not significant; RGS, regulator of G-protein signaling; sf, superfolder; Sst2, supersensitive 2; Ste, sterile; tdTomato, tandem dimer tomato.

allowing accumulation all over the membrane) [50,51] or had a constitutively active strong endocytosis signal (Ste2[NPF], where the GPFAD endocytosis motif is mutated to NPFAD, yielding a highly polarized distribution with a bias toward the mother-bud neck) [52] (Fig 7A and 7B). Because endocytosis is needed for Ste2 degradation, Ste2[7XR-GPAAD] was more abundant than Ste2 or Ste2[NPF] (Fig 7C), and in halo assays, cells expressing Ste2[7XR-GPAAD] were slightly more sensitive to pheromone (Fig 7D). We assume the small effect on halo size is due to ratiometric accommodation to the increased receptor level because cells with compromised ratiometric sensing displayed a much greater increase in global sensitivity when they expressed Ste2[7XR-GPAAD] (Fig 7E).

To quantify the accuracy of initial clustering, we recorded the location of Bem1 clusters among all cells that were born immediately adjacent to a G1 cell of opposite mating type. Despite the dramatic difference in receptor distribution (Fig 7B), cells with Ste2[NPF] or Ste2[7XR-GPAAD] were comparable to wild-type cells at orienting their initial clusters towards partners (Fig 7F). This finding suggests that yeast can correct for variations in receptor density. However, the mutants in which Sst2 was replaced with hsRGS4 exhibited a polarization accuracy that was dramatically affected by the receptor distribution (Fig 7G). In these mutants, initial polarization was accurate only in cells with uniformly distributed receptors and became random in cells with highly polarized receptors. We infer that yeast cells use Sst2-dependent local ratiometric sensing of receptor occupancy to extract accurate information from the pheromone gradient despite having nonuniform receptor density.

## Ratiometric sensing amplifies the pheromone gradient

In addition to protecting cells from being misled by uneven receptor density, ratiometric sensing could, in principle, confer a benefit even in cells that had uniformly distributed receptors. This is because a gradient of pheromone would generate both a gradient in the concentration of ligand-bound receptors and an opposing gradient in the concentration of unoccupied receptors (Fig 8A). Consider a gradient of active receptor rising from left to right across the cell, with 50% active receptor in between. If we compare ratiometric and nonratiometric sensing models matched so that the G-protein deactivation rate in both models is equal when 50% of the receptor is bound to ligand, then the active G-protein concentration on the right side will be higher for the ratiometric model because the inactivation rate (mediated by inactive receptor, which is <50% on this side) is lower. Similarly, the active G-protein concentration on the left side will be lower for the ratiometric model because the inactivation rate is higher. Thus, the difference in active G-protein concentration between the two ends of the cell will always be larger in the ratiometric model.

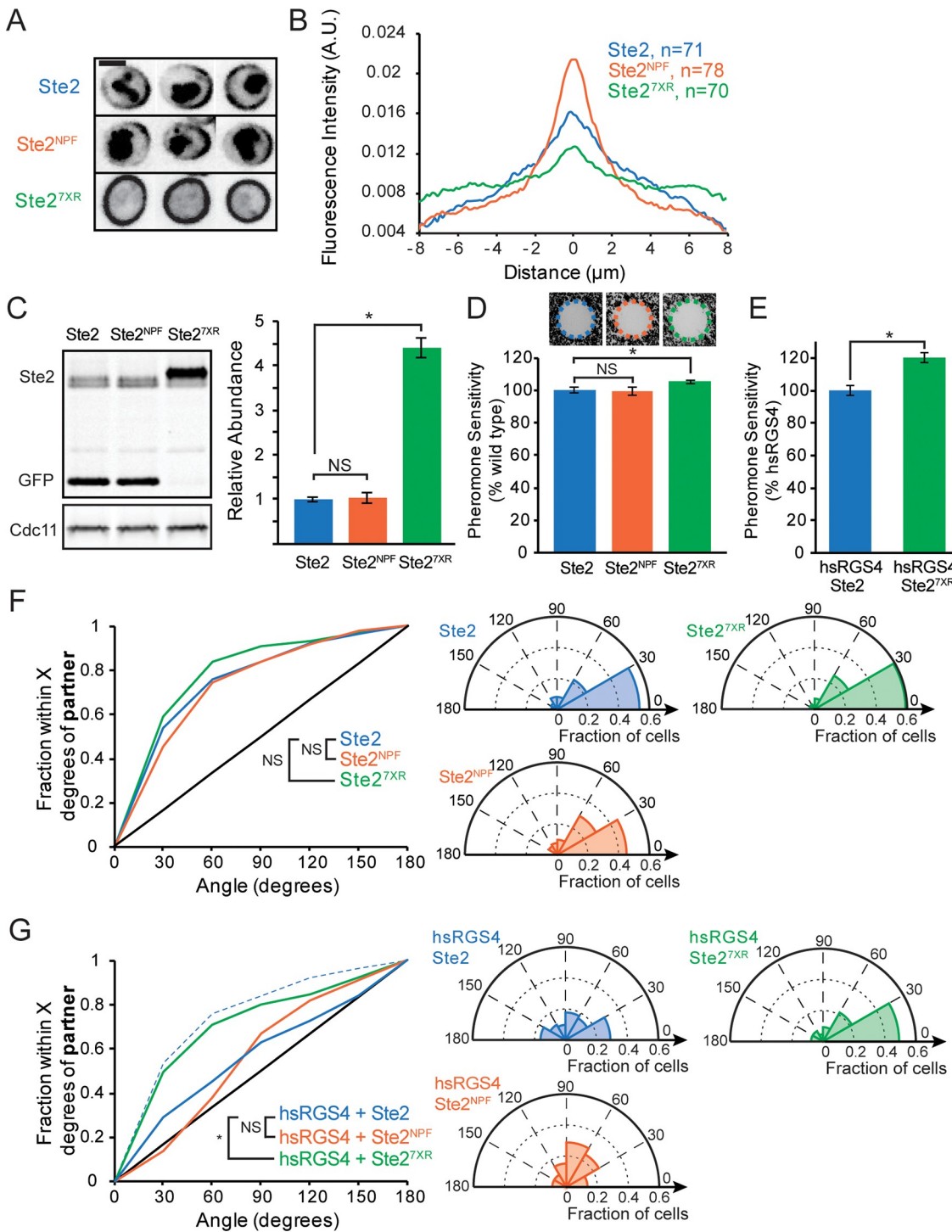

**Fig 7. Ratiometric sensing makes gradient detection robust to changes in receptor distribution.** (**A**) Single-plane inverted images of Ste2–sfGFP (top), Ste2$^{NPF}$–sfGFP (middle), and Ste2$^{7XR-GPAAD}$–sfGFP (bottom) in representative G1 cells. Ste2$^{7XR-GPAAD}$–sfGFP images were scaled differently to compensate for increased abundance. (**B**) Average Ste2 membrane distribution, quantified as in Fig 5D, in G1 cells with Ste2–sfGFP (blue), Ste2$^{NPF}$–sfGFP (orange), and Ste2$^{7XR-GPAAD}$–sfGFP (green). (**C**) Ste2–sfGFP abundance. Left: representative western blot (full uncropped blot available in S5 Fig). α-GFP antibodies label two bands—full-length Ste2–sfGFP and vacuolar sfGFP (note absence of vacuole signal for Ste2$^{7XR-GPAAD}$). Right: quantification of full-length Ste2 abundance ($n = 3$ biological replicates, normalized to the average abundance of wild-type Ste2). (**D**) Halo assay for global pheromone sensitivity of cells with wild-type Ste2 (blue), Ste2$^{NPF}$ (orange), and Ste2$^{7XR-GPAAD}$ (green). Top: images of representative halos. Bottom: quantification of halo diameter ($n = 12$, 3 technical replicates, normalized to the average wild-type halo diameter; *$t$ test, $p < 0.05$). (**E**) Halo assay as

in (D) for cells with hsRGS4×2 in place of *SST2*, with wild-type Ste2 (blue) and Ste2$^{7XR-GPAAD}$ (green). (**F**) Left: Cumulative distribution of initial Bem1 cluster orientation relative to the nearest potential mating partner for MAT**a** cells born immediately adjacent to a MATα cell in G1. Cells with wild-type Ste2 (blue, $n = 222$), Ste2$^{NPF}$ (orange, $n = 93$, NS), or Ste2$^{7XR-GPAAD}$ (green, $n = 148$, NS). Right: polar histograms of the same data. (**G**) Left: Cumulative distribution of initial Bem1 cluster orientation as in (E) for cells with hsRGS4×2 in place of *SST2*, with wild-type Ste2 (blue, $n = 62$), Ste2$^{NPF}$ (orange, $n = 66$, NS), or Ste2$^{7XR-GPAAD}$ (green, $n = 65$, *two-sample KS test, $p < 0.05$). Note: hsRGS4×2 + Ste2$^{7XR-GPAAD}$ (green) is not significantly different from wild type + *SST2* (blue dashed line). Right: polar histograms of the same data. Strains: DLY20713, DLY20715, DLY21705 (A, B), DLY21203, DLY21206, DLY21704 (C), DLY8993, DLY21205, DLY21206 (D), DLY23623, DLY23624 (E), DLY12943, DLY22058, DLY22397 (F), DLY22520, DLY22570, DLY22606 (G). A.U., arbitrary unit; Bem1, bud emergence 1; Cdc, cell division cycle; GFP, green fluorescent protein; GPAAD, GPFAD to GPAAD mutation; hsRGS4, *Homo sapiens* regulator of G-protein signaling 4; KS, Kolmogorov–Smirnov; MAT, mating type; NPF, GPFAD to NPFAD mutation; NS, not significant; RGS, regulator of G-protein signaling; sf, superfolder; Sst2, supersensitive 2; Ste, sterile; 7XR, 7 lysine-to-arginine mutations.

To understand how large the benefits of ratiometric sensing might be, we developed a simple model assuming that local G-protein activity reaches steady state (S1 Text). As an example, we calculated the gradients of active G protein across the cell that would result from linear gradients of receptor activity centered at 50% active receptor so that both models have the same concentration of G protein at the cell middle (Fig 8B). As expected, steeper gradients of receptor activity produced steeper gradients of active G protein. The crucial parameter controlling the shape of the G-protein gradients in these conditions is the ratio between the G-protein inactivation and activation rate constants (β in Fig 8A and 8B). Raising this ratio lowers the active G-protein concentration and vice versa; when the ratio is 1, the G-protein gradient equals the receptor gradient for the ratiometric model. Fig 8B illustrates the gradients of active G protein predicted on the assumption of ratiometric or nonratiometric sensing.

To explore a wider range of gradients and values of β, we measured the difference in active G-protein concentration (ΔG*) across 5 μm (representing the width of the cell) and calculated a Signal Ratio (SR) as the ratio of this difference between the models: SR = ΔG*$_{ratiometric}$/ ΔG*$_{nonratiometric}$. Thus, SR values >1 indicate a greater signal for the ratiometric model. For shallow gradients, the ratiometric model outperforms the nonratiometric model by a factor of 2. With steeper gradients, the SR varies with β and can exceed 2 (Fig 8C).

The example gradients shown in Fig 8B show that the ratiometric and nonratiometric models differ not only in the magnitude of the difference between the G-protein concentrations at the endpoints (ΔG*), but also in the shape of the G-protein gradients. Notably, whenever β > 1, the ratiometric model produces gradients that are convex, meaning that the G-protein concentration rises more steeply on the right (up-gradient side) than on the left. In contrast, gradients in the nonratiometric model are always concave, rising more steeply on the left.

Convexity in the gradient profile may confer an advantage in terms of deciding where to place the polarity site because a higher fraction of the active G protein is concentrated near that pole, as illustrated in Fig 8D. This may enable the cell to make a more robust choice of polarity site in the presence of noise. With a concave profile, cells may not be able to distinguish where the G-protein concentration is highest because the profile is flatter at the high end. For a fixed net difference in active G-protein concentration between the cell ends, a convex gradient would concentrate more G protein up-gradient than a concave gradient, and the high-concentration zone would be narrower (shaded areas in Fig 8D). Thus, in addition to translating the same receptor gradient into a G-protein gradient with a larger difference between the cell ends, ratiometric sensing may further facilitate accurate orientation by sculpting the gradient shape to better delineate the up-gradient pole. Both features should improve the signal-to-noise ratio for gradient detection.

The discussion above neglected diffusion. To examine the benefit of ratiometric sensing without neglecting diffusion, we conducted particle-based simulations with uniform receptor

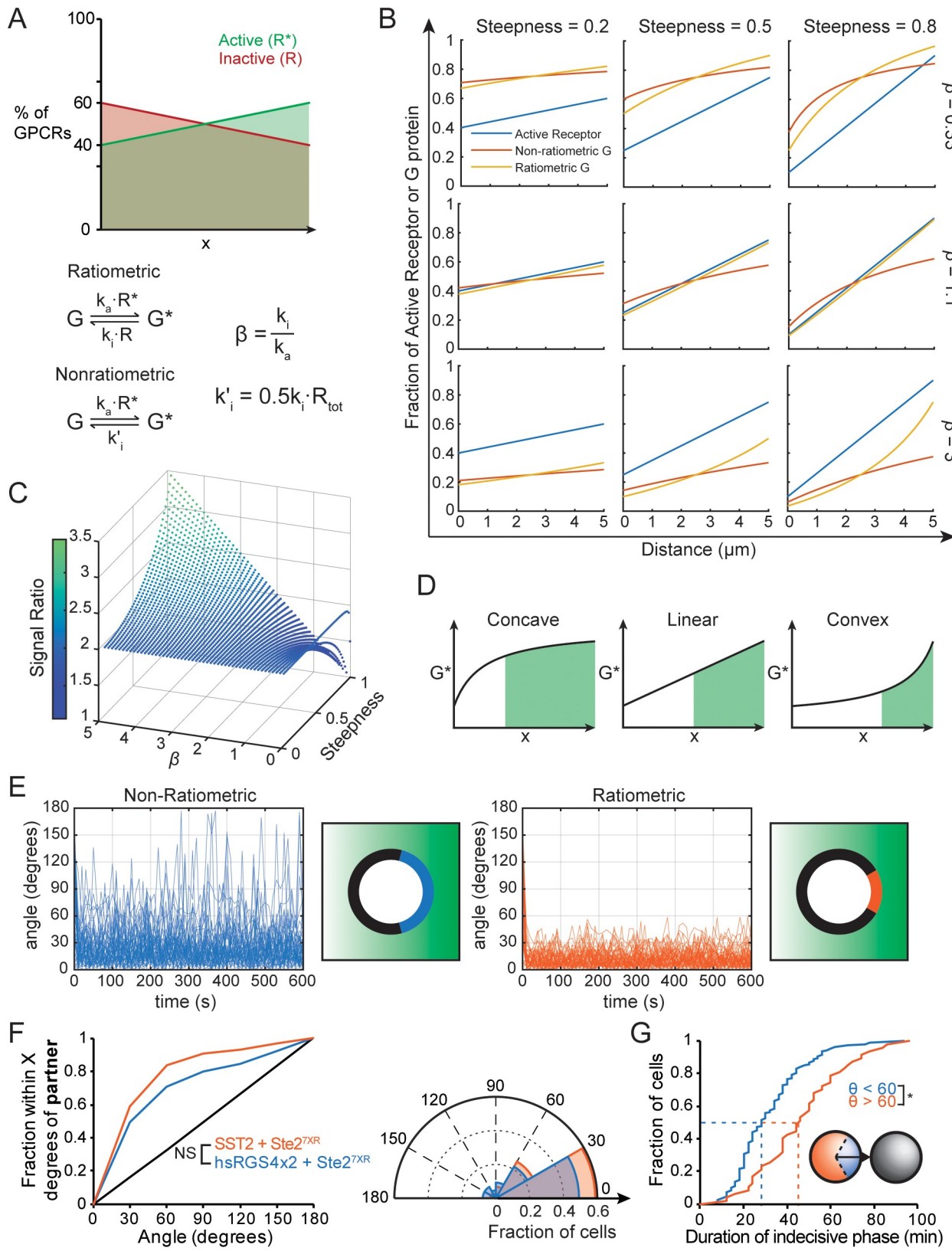

**Fig 8. Ratiometric sensing amplifies the gradient signal and improves accuracy even when receptors are distributed uniformly.** (**A**) When receptor distribution is uniform, a gradient of active receptors automatically implies an opposing gradient of inactive receptors (top). Bottom: G-

protein activation and inactivation rates in ratiometric versus nonratiometric models, criterion for matching inactivation rates in the two models, and definition of β. (**B**) Predicted gradient profiles for active G protein in ratiometric (yellow) and nonratiometric (orange) models illustrated for receptor activity gradients (blue) of differing steepness and for different values of β. Steepness of the receptor gradient is the slope normalized to the maximum slope of a linear gradient with 0% active receptors at left and 100% active receptors at right. (**C**) Plot of the SR (vertical axis and color bar) for different gradient steepness and β. The SR refers to the difference in active G-protein concentration between the two ends of the gradient predicted by the ratiometric model divided by that predicted by the nonratiometric model, assuming steady state. (**D**) For gradients that exhibit the same signal, the shape of the gradient can affect the accuracy with which a cell would pick the right site for polarization. Three gradient shapes are illustrated for the same signal, and the shaded region indicates the part of the gradient where the active G-protein concentration is above the average. (**E**) Simulations with uniform receptor density. The ratiometric (orange) and nonratiometric (blue) models were simulated as in Fig 6B. (**F**) Left: Cumulative distribution of initial Bem1 cluster orientation relative to the nearest potential mating partner for MAT**a** cells born immediately adjacent to a MATα cell in G1. Cells with Ste2$^{7XR-GPAAD}$ (uniform receptors) and either wild-type Sst2 (orange, $n = 148$), or hsRGS4×2 (blue, $n = 65$). Right: polar histograms of the same data. (**G**) Cumulative distribution of the duration of the indecisive phase in second-born cells, plotted separately for cells in which the initial cluster formed within 60˚ of the mating partner (θ < 60˚, blue, $n = 106$), and cells in which the initial cluster formed greater than 60˚ from the partner (θ > 60˚, orange, $n = 47$) (*two-sample KS test, $p < 0.05$). Inset: diagram displaying the two groups of cells. Code for Fig 8B and 8C is available at https://github.com/DebrajGhose/Ratiometric-GPCR-signaling-enables-directional-sensing-in-yeast. Code and key data for Fig 8E are available at https://github.com/mikepab/ratiometric-gpcr-particle-sims. Strains: DLY22397, DLY22606 (F), DLY12943, DLY7593 (G). Bem1, bud emergence 1; GPCR, G-protein–coupled receptor; GPAAD, GPFAD to GPAAD mutation; hsRGS4, *Homo sapiens* regulator of G-protein signaling 4; KS, Kolmogorov–Smirnov; MAT, mating type; NS, not significant; RGS, regulator of G-protein signaling; SR, Signal Ratio; Sst2, supersensitive 2; Ste, sterile; 7XR, 7 lysine-to-arginine mutations.

density. As predicted, the ratiometric model reduced the noise in the direction of the gradient as compared to the nonratiometric model (Fig 8E). This benefit was preserved with receptor diffusion (S4A Fig), and ratiometric sensing outperformed nonratiometric sensing even with as few as 1,000 total receptors (S4B Fig). Thus, ratiometric sensing can, in principle, provide a significant benefit even to cells with uniform receptors.

In yeast cells that had more uniform receptor distribution because of mutations blocking receptor endocytosis, the accuracy of initial polarization was also improved by ratiometric-sensing–competent Sst2 as compared to nonratiometric hsRGS4, although the difference was not statistically significant for the number of cells analyzed (Fig 8F).

Because cells undergo an indecisive phase after initial clustering of polarity factors and can successfully mate even when initial clustering is poorly oriented, one might question whether the accuracy of initial cluster orientation is important for subsequent mating. To address that question, we compared the duration of the indecisive phase between cells that formed their initial clusters within 60˚ of their partners and those whose initial clusters were less well-oriented (Fig 8G). The indecisive phase duration was significantly shorter for well-oriented than misoriented cells (median 28 min versus 46 min) (Fig 8G). Thus, gradient sensing before polarity cluster formation can shorten the search for a partner.

## Discussion

### Initial polarity cluster location is surprisingly accurate

The rapid diffusion of peptide pheromones and the small size of the yeast cell led to the expectation that there would be only a small difference in pheromone concentration between the up- and down-gradient sides of the cell. This poses a fundamental difficulty in extracting accurate directional information in the face of molecular noise [20]. Indeed, cells responding to a 0.5 nM/μm pheromone gradient were reported to orient initial polarity clusters almost at random [36]. Moreover, the polarity circuit in yeast contains strong positive feedback [15,53], which allows cells to polarize in random directions when treated with uniform pheromone concentrations [24,45]. This would be expected to enable noise-driven polarization in random directions in cells exposed to shallow gradients [54]. Making matters even worse, we documented significant receptor asymmetry, with (on average) 3-fold more concentrated receptors on one side of the cell than the other. This creates a receptor density gradient that is significantly steeper than the assumed pheromone gradient. Because the receptor density gradient is

randomly oriented with respect to the mating partner, this poses a serious hurdle in accurate gradient detection.

Despite the difficulties enumerated above, the location of initial polarity factor clustering in mating mixtures was highly nonrandom and surprisingly accurate, with more than 50% of cells clustering within 30° of the correct direction and less than 5% of cells clustering in the opposite segment (a random process would have 17% of cells polarizing in each segment). This accuracy is all the more remarkable because initial clustering of polarity factors occurred within 5.1 ± 2.8 min from cell birth. Given the slow timescale of yeast pheromone receptor binding and dissociation [55–57], this rapid polarization may allow cells to take advantage of pre-equilibrium sensing and signaling [58], a proposed mechanism in which directionality is inferred from the rates of pheromone binding rather than steady-state distributions. However, that mechanism would not explain how accurate orientation could be achieved in the face of highly uneven receptor density. Thus, our findings suggest that yeast cells possess unappreciated mechanisms to overcome the difficulties in accurate gradient detection discussed above.

## Orientation accuracy is enhanced by ratiometric sensing

One way to avoid being misled by an asymmetric receptor distribution would be to compare the local ratio of occupied and unoccupied receptors, rather than simply the density of occupied receptors, across the cell surface. An elegant mechanism to extract such information was proposed by [37]. Because the RGS protein Sst2 binds to unoccupied receptors [50], those receptors promote GTP hydrolysis by Gα. Conversely, occupied receptors catalyze GTP-loading by Gα. Thus, the net level of GTP–Gα reflects the fraction (and not the number) of occupied receptors on the cell [37]. For this mechanism to promote local ratiometric sensing, it requires additionally that a pheromone-bound receptor diffuse slowly relative to its lifetime at the surface (approximately 10 min) [29] so that information about where receptors were when they bound to pheromone is not lost. We found that receptors do indeed diffuse very slowly (D < 0.0005 μm$^2$/s) at the yeast plasma membrane. Moreover, ratiometric gradient sensing requires that the ratio of active to inactive receptors is measured locally rather than globally. Simulations with realistic numbers of receptors and G proteins demonstrated that this mechanism has the potential to extract unbiased information about the pheromone gradient even in the face of uneven receptor density.

We found that when RGS function was delocalized by replacing Sst2 (which binds unoccupied receptors) with an equivalently active amount of hsRGS4 (which binds the plasma membrane), the accuracy of initial polarity clustering was severely compromised. Instead of polarizing towards potential partners, these cells assembled polarity clusters at regions where receptors were concentrated. Thus, abrogating the Sst2-based ratiometric sensing mechanism allowed cells to be misled by the asymmetric receptor distribution. Accurate orientation could be restored to these cells by making receptor distribution more uniform. In sum, our findings suggest that local ratiometric sensing compensates for uneven receptor distribution and allows more accurate polarization towards mating partners.

An additional benefit of ratiometric sensing in terms of gradient detection is that this mechanism exploits both the gradient in ligand-bound receptors and the complementary gradient in unoccupied receptors to sharpen the downstream G-protein gradient. This feature would be beneficial even in cells with uniform receptor density. Whereas for yeast, the main function of ratiometric sensing appears to be to correct for the uneven receptor distribution, we speculate that in other systems in which receptors are distributed more uniformly, ratiometric sensing would still be beneficial as a gradient amplification mechanism. Interactions between mammalian RGS

proteins and specific GPCRs, analogous to the Sst2–Ste2 interaction in yeast, have been identified in many contexts [59–61]. Indeed, one such interaction involved CXC chemokine receptor 2 (CXCR2), which mediates chemotactic responses in leukocytes [62]. It will be interesting to determine whether other GPCRs exploit ratiometric sensing to sharpen gradient detection.

### Error correction following initial clustering of polarity factors

Although initial polarity clusters were biased to occur near potential mating partners, the process was error-prone, and about 50% of cells failed to orient initial polarity within 30˚ of the correct direction. Nevertheless, these cells did eventually polarize towards partners and mate successfully, indicating the presence of a potent error correction mechanism. We found that after initial clustering, polarity factor clusters relocated erratically during an "indecisive phase" of variable duration (45 min ± 23 min). Even cells that had correctly assembled initial polarity clusters close to mating partners exhibited an indecisive phase, although of shorter duration. During this phase, clusters fluctuated in intensity (concentration of polarity factors in the cluster), extent (broader versus more focused clusters), location, and number (transiently showing no cluster or 2–3 clusters instead of a single cluster).

At the end of the indecisive phase, cells developed strong and stable polarity sites correctly oriented towards their partners, entering a "committed phase." A similar stabilization of polarity clusters was noted in cells exposed to an artificial pheromone gradient [36]. Previous studies used the term commitment to indicate a change in MAPK activity [44] or mating pathway gene induction [63] occurring 15–30 min prior to fusion, and it seems probable that all of these changes are linked and that an increase in MAPK activity promotes commitment.

The increase in MAPK activity may stem from integration of the pheromone signal over time until a threshold is reached, triggering a rapid increase in MAPK activity. In this scenario, the transient and weak indecisive phase polarity clusters may have no functional role. Alternatively, even weak polarity clusters may act both as sources of pheromone secretion and locations of pheromone sensing, as suggested by the exploratory polarization hypothesis (see Introduction). That hypothesis was based on the behavior of polarity clusters in cells that had been genetically manipulated to induce maximal MAPK activity [26]. In such cells, a strong polarity cluster moves gradually around the cell cortex, surrounded by a receptor- and G-protein–enriched sensitized zone. It remains unclear whether the weaker and much more labile indecisive phase polarity clusters can similarly engage in exploratory polarization. Nevertheless, the exploratory polarization strategy, like the related "speed dating" strategy proposed for fission yeast cells [34,35], has several attractive features. In particular, this strategy converts a very difficult problem (extracting directional information from shallow and noisy pheromone gradients) into a much easier one (detecting a sharp temporal increase in local pheromone level).

## Materials and methods

### Yeast strains and plasmids

Yeast strains used in this study are listed in Table 1. Standard yeast molecular and genetic procedures were used to generate the strains. All strains are in the YEF473 background (*his3-Δ200 leu2-Δ1 lys2-801 trp1-Δ63 ura3-52*) [64]. The following alleles were previously described: Bem1–GFP [17], Bem1–tdTomato and Spa2–mCherry [65], GFP–Sec4 [24], STE2$^{7XR-GPAAD}$ [26], Ste7$_{1–33}$–nuclear localization sequence (NLS)–NLS–mCherry [46], and hsRGS4–CFP [37].

Spa2–GFP tagged at the endogenous locus was generated by the PCR-based method using pFA6–GFP(S65T)–HIS3MX6 as template [66].

**Table 1. Yeast strains and genotypes.**

| Strain | Relevant Genotype |
|---|---|
| DLY7593 | *MATα ura3:BEM1–GFP:URA3* |
| DLY7594 | *MATa ura3:BEM1–GFP:URA3* |
| DLY8156 | *MATα* |
| DLY8993 | *MATa bar1ΔURA3* |
| DLY9070 | *MATα BEM1–GFP:LEU2* |
| DLY12943 | *MATa BEM1–tdTomato:HIS3* |
| DLY13771 | *MATa BEM1–tdTomato:HIS3, GFP–SEC4:URA3* |
| DLY20713 | *MATa SPA2–mCherry:hyg$^R$, STE2–sfGFP:URA3* |
| DLY20715 | *MATa SPA2–mCherry:hyg$^R$, STE$^{NPF}$–sfGFP:URA3* |
| DLY21070 | *MATα BEM1–tdTomato:HIS3* |
| DLY21203 | *MATa SPA2–mCherry:hyg$^R$, STE2–sfGFP:URA3, bar1ΔURA3* |
| DLY21205 | *MATa SPA2–mCherry:kan$^R$, STE$^{7XR-GPAAD}$–sfGFP:URA3, bar1ΔURA3* |
| DLY21206 | *MATa SPA2–mCherry:kan$^R$, STE$^{NPF}$–sfGFP:URA3, bar1ΔURA3* |
| DLY21295 | *MATa STE2$^{7XR-GPAAD}$:URA3* |
| DLY21301 | *MATa STE2$^{NPF}$:URA3* |
| DLY21379 | *MATa BEM1–tdTomato:HIS3, SPA2-GFP:HIS3* |
| DLY21704 | *MATa SPA2–mCherry:kan$^R$, STE2$^{7XR-GPAAD}$–sfGFP:LEU2:URA3, bar1ΔURA3* |
| DLY21705 | *MATa SPA2–mCherry:hyg$^R$, STE2$^{7XR-GPAAD}$–sfGFP:LEU2:URA3* |
| DLY22058 | *MATa ura3:BEM1–GFP:URA3, STE2$^{NPF}$:URA3* |
| DLY22243 | *MATa BEM1-tdTomato:HIS3, STE2–sfGFP:URA3* |
| DLY22259 | *MATa SPA2–GFP:HIS3, ura3:Ste7$_{1–33}$–NLS–NLS–mCherry:URA3* |
| DLY22318 | *MATa BEM1–GFP:LEU2, SST2:hsRGS4–CFP:kan$^R$* |
| DLY22321 | *MATa BEM1–GFP:LEU2* |
| DLY22340 | *MATα BEM1–tdTomato:HIS3* |
| DLY22397 | *MATa BEM1–GFP:LEU2, STE2$^{7XR-GPAAD}$:URA3* |
| DLY22520 | *MATa BEM1–GFP:LEU2, SST2:hsRGS4–CFP:kanR, ura3:P$_{SST2}$–hsRGS4–CFP:URA3* |
| DLY22570 | *MATa BEM1–GFP:LEU2, STE2$^{NPF}$:URA3, SST2:hsRGS4–CFP:kan$^R$, ura3: P$_{SST2}$–hsRGS4–CFP:URA3* |
| DLY22606 | *MATa BEM1–GFP:LEU2, STE2$^{7XR-GPAAD}$:URA3, SST2:hsRGS4–CFP:kan$^R$, ura3: P$_{SST2}$–hsRGS4–CFP: URA3* |
| DLY22628 | *MATa BEM1–tdTomato:HIS3, STE2–sfGFP:URA3, SST2:hsRGS4–CFP:kan$^R$, ura3: P$_{SST2}$–hsRGS4–CFP: URA3* |
| DLY22883 | *MATα BEM1–tdTomato:HIS3, leu2:P$_{MFα1}$–sfGFP:LEU2* |
| DLY22764 | *MATa SPA2–GFP:HIS3, ura3:Ste7$_{1–33}$–NLS–NLS–mCherry:URA3, ste5:P$_{GAL1}$–STE5–CTM:P$_{ADH1}$– GAL4BD–hER–VP16:LEU2* |
| DLY22928 | *MATa/α BEM1–tdTomato:HIS3/BEM1, cdc12-6/CDC12, P$_{TEF1}$–GFP:LEU2/leu2* |
| DLY23623 | *MATa BEM1–GFP:LEU2, SST2:hsRGS4–CFP:kan$^R$, ura3: P$_{SST2}$–hsRGS4–CFP:URA3, bar1ΔURA3* |
| DLY23624 | *MATa BEM1–GFP:LEU2, STE2$^{7XR-GPAAD}$:URA3, SST2:hsRGS4–CFP:kan$^R$, ura3: P$_{SST2}$–hsRGS4–CFP: URA3, bar1ΔURA3* |

**Abbreviations:** Bem1, bud emergence 1; Cdc, cell division control; CFP, cyan fluorescent protein; GFP, green fluorescent protein; GPAAD, GPFAD to GPAAD mutation; hsRGS4, *Homo sapiens* regulator of G-protein signaling 4; MAT, mating type; NPF, GPFAD to NPFAD mutation; P$_{gal1}$, galactose metabolism 1 promoter; RGS, regulator of G-protein signaling; sf, superfolder; Spa2, spindle pole antigen 2; Sst2, supersensitive 2; Ste, sterile; tdTomato, tandem dimer tomato; 7XR, 7 lysine-to-arginine mutations.

To express Ste2–sfGFP, sfGFP was amplified by PCR using pFA6a–link–yoSuperfol-derGFP–KAN (Addgene plasmid 44901; Cambridge, MA, USA) as template, with primers that added *Not*I sites at the ends. This was used to generate DLB4295, a plasmid with a pRS306 backbone [67] and a C-terminal piece of the *STE2* ORF (bases 600–1,296) fused to sfGFP and

followed by 198 bp of the *STE2* 3′-UTR. Digestion at the unique *Cla*I site in *STE2* targets integration of this plasmid to the endogenous *STE2* locus, tagging full-length Ste2 with sfGFP at the C-terminus.

Similar plasmids were used to express Ste2$^{7XR-GPAAD}$–sfGFP (DLB4296) and Ste2$^{NPF}$–sfGFP (DLB4297) at the endogenous *STE2* locus. Ste2$^{NPF}$ was generated by first amplifying a fragment of *STE2* using primers that introduced a GGA → AAT mutation (G$_{392}$N substitution) [52] and cloning the fragment back into *STE2*.

Ste7$_{1-33}$–NLS–NLS–mCherry was integrated at *ura3* using pED45 (pRS306–P$_{RPS2}$–Ste7$_{1-33}$–*NLS–NLS–mCherry*) as described [46].

To compromise ratiometric sensing by Sst2, we replaced the endogenous *SST2* with hsRGS4–CFP using A550 (pRS406–K–*hsRGS4–CFP*) as described [37]. Because this was insufficient to restore wild-type pheromone sensitivity in our strain background, P$_{SST2}$–*hsRGS4–CFP* was amplified by PCR and cloned into pRS306 using *Xba*I to generate DLB4414. Digestion with *Stu*I was then used to target integration of a second copy of hsRGS4–CFP at *URA3*.

To make the *MFα1* reporter, the *MFα1* promoter (506 base pairs upstream of the ATG) was amplified with primers that added *Apa*I and *Hind*III sites and cloned upstream of a reporter protein with the first 28 residues of Psr1 fused to GFP, followed by the *ADH1* 3′-UTR, in a plasmid with a pRS305 (*LEU2*) backbone [67]. The Psr1$_{1-28}$–GFP reporter was replaced with sfGFP, which was cloned from pFA6a–link–yoSuperfolderGFP–KAN (Addgene plasmid 44901). Digestion at the *Ppu*MI in the *LEU2* sequence was used to target integration at *leu2*. To make the *TEF1* reporter, the *TEF1* promoter (464 base pairs upstream of the ATG) was amplified with primers that added *Bgl*II and *Pac*I sites and cloned into pFA6a–TRP1–pGAL–GFP, replacing the GAL promoter upstream of GFP [66]. pGAL–GFP was amplified with primers that added *Bgl*II and *Not*I sites and cloned a plasmid with a pRS305 (*LEU2*) backbone [67]. Digestion at the *Ppu*MI in the *LEU2* sequence was used to target integration at *leu2*.

To induce MAPK activation without adding pheromone, we generated a plasmid, DLB4239 (pRS305-STE5$_{5'UTR}$–STE5$_{3'UTR}$–P$_{ADH1}$–GAL4BD–hER–VP16–P$_{GAL1}$–STE5–CTM), that can be used to replace the endogenous *STE5* locus with two genes: (i) a hybrid transcription factor that activates Gal4 target genes in response to estradiol (GAL4BD–hER–VP16) [68], and (ii) a *GAL1* promoter driving expression of a membrane-targeted version of Ste5 (P$_{GAL1}$–STE5–CTM) [8]. Addition of estradiol activates the transcription of membrane-targeted Ste5, which leads to activation of the mating MAPKs. The plasmid has a pRS305 (*LEU2*) backbone and contains regions of the *STE5* 5′- and 3′-UTRs upstream of the hybrid transcription factor. DLY4239 was digested with *Pac*I, which cuts between the *STE5* 5′- and 3′-UTR regions to replace endogenous *STE5* with the two genes.

## Live-cell microscopy

Cells were grown to mid-log phase (OD$_{600}$ ≈ 0.4) overnight at 30˚C in complete synthetic medium (CSM; MP Biomedicals, LLC, Solon, OH, USA) with 2% dextrose (Macron, Center Valley, PA, USA). Cultures were diluted to OD$_{600}$ = 0.1. For mating mixtures, the relevant strains were mixed 1:1 immediately before mounting on slabs. Cells were mounted on CSM slabs with 2% dextrose solidified with 2% agarose (Hoefer, Holliston, MA, USA), which were then sealed with petroleum jelly. For Ste5–CTM MAPK induction, slabs also contained 20 nM β-estradiol (Sigma-Aldrich, St. Louis, MO, USA). Cells were imaged in a temperature-controlled chamber set to 30˚C.

Images were acquired with an Andor Revolution XD spinning disk confocal microscope (Andor Technology, Concord, MA, USA) with a CSU-X1 5,000-rpm confocal scanner unit (Yokogawa, Tokyo, Japan), and a UPLSAPO 100×/1.4 oil-immersion objective (Olympus,

Tokyo, Japan), controlled by MetaMorph software (Molecular Devices, San Jose, CA, USA). Images were captured by an iXon3 897 EM-CCD camera with 1.2× auxiliary magnification (Andor Technology).

For high-resolution images of Ste2–sfGFP, Ste2$^{NPF}$–sfGFP, and Ste2$^{7XR-GPAAD}$–sfGFP, z-stacks with 47 planes were acquired at 0.14-μm intervals. The laser power was set to 30% maximal output, EM gain was set to 200, and the exposure for the 488 nm laser was set to 250 ms. For all other microscopy, z-stacks with 15 images were acquired at 0.5-μm z-steps every 2 min; laser power was set to 10% maximal output for the relevant 488-nm, 561-nm, or 445-nm lasers; EM gain was set to 200; and the exposure time was 200 ms.

All fluorescent images were denoised using the Hybrid 3D Median Filter plugin for ImageJ, developed by Christopher Philip Mauer and Vytas Bindokas.

## Analysis of the timing of cell cycle and mating events

Bud emergence was scored using DIC confocal images. Cytokinesis was recorded as the first time point when a strong Bem1 signal was visible at the neck. Initial clustering was recorded as the first time point after cytokinesis when a Bem1 cluster was clearly visible and distinguishable from background noise. Polarization commitment was recorded as the time point when the Bem1 patch reached its final stable location and increased in intensity. If the patch appeared at the correct location but then transiently moved to a new location before returning, polarization was recorded as the time point when the patch returned. Fusion was recorded as the time when cytoplasmic mixing of different color probes became detectable.

## Analysis of polarity factor clustering

To quantify the degree of clustering of the polarity probes Spa2–mCherry, Bem1–tdTomato, and Bem1–GFP, we calculated a "deviation from uniformity" metric from maximum projections of fluorescent z-stack images. Deviation from uniformity, referred to here as clustering parameter (CP), compares the cumulative distribution of pixel intensities in an actual cell with that in a hypothetical cell with the same range of pixel intensities that are uniformly distributed; i.e., CP measures how different the pixel intensity distribution is from a uniform distribution, which reflects the degree to which the signals are clustered.

An elliptical region of interest (ROI) was drawn around each cell at each time point. Raw pixel intensities (p) within each ROI were normalized to a minimum of 0 and maximum of 1:

$$i = \left( \frac{p - p_{min}}{p_{max} - p_{min}} \right).$$

A cumulative distribution (D) of pixel intensities (i) within the cell is then calculated as

$$D_i = \frac{(no.of\ pixels\ with\ intensity < i)}{(total\ no.of\ pixels)}.$$

For a cell with uniformly distributed pixel intensities, the cumulative distribution (U) is

$$U_i \approx i.$$

500 uniformly spaced i-values from 0 to 1 were indexed in ascending order as $n$ = 1, 2, 3, ..., 500. The deviation from uniformity metric (CP) was calculated as

$$CP = 2 \cdot \sum_{n=1}^{500} (D_i - U_i).$$

CP approaches a maximum of 1 when a small fraction of pixels exhibit near the maximum intensity while most pixels are clustered near the minimum intensity, as seen in a highly polarized cell. CP is sensitive to the size of the patch and the distribution of intensities within the patch—a small patch with sharp edges yields a high CP, while a broad patch with graded edges yields a low CP. As a result, CP is a sensitive indicator of the transition between the indecisive and committed phases.

CP was measured using a MATLAB (The MathWorks, Natick, MA, USA)-based graphical user interface called ROI_TOI_QUANT_V8, developed by Denis Tsygankov.

### Estimation of diffusion constants using FRAP

Transit of pheromones and pheromone receptors through the secretory pathway is rapid (5–10 min) [69,70]. In the presence of α-factor, Ste2 is then endocytosed on a 10-min timescale and delivered to the vacuole for degradation [27–30]. Because GFP maturation occurs on a 30-min timescale [71,72], much of the GFP-tagged receptor at the cell surface is not yet fluorescent. Moreover, the GFP moiety survives intact in the vacuole following receptor degradation, yielding a high vacuolar fluorescence signal. To resolve these issues, we used a nonendocytosable version of Ste2, Ste2$^{7XR-GPAAD}$ [50,51], and tagged this protein with sfGFP, which matures on a 6-min timescale [73]. In budding cells, all new secretion is targeted to the bud [38]. Thus, we avoided fluorescence recovery via delivery of new receptors by bleaching zones in the mother segment of budded cells.

We performed FRAP with a DeltaVision deconvolution microscope (GE Healthcare Life Sciences, Chicago, IL, USA) on cells expressing either Ste2$^{7XR-GPAAD}$–sfGFP or a nonendocytosable v-SNARE (suppressor of the null allele of CAP 2 [Snc2]$^{V39A,M42A}$) tagged with GFP. We bleached the mother portions of budded cells, avoiding the bud and the distal pole of the mother to ensure that most of the fluorescence recovery was due to diffusion and not vesicle trafficking. Bleaching was performed with a 488-nm laser set to 20% power for 200 ms. The fluorescence intensity in the bleached region was then tracked over time at 15-s intervals for 10 min. The signal in each cell's bleached region was normalized by dividing it by the total fluorescence of the mother cell at each time point to account for photobleaching during image acquisition. The recovery rate constant was then estimated by fitting an exponential function to the time-series data, and the Soumpasis equation was used to derive the diffusion constant from the recovery rate constant [74]. Image analysis and other calculations were carried out in MATLAB 2018b.

### Analysis of initial polarity cluster orientation

Initial orientation was measured at the time of initial clustering. For orientation relative to the partner, we measured the angle between the line from the center of the cell being scored to the centroid of the initial cluster and a line from the cell center to the closest surface of the nearest G1 cell of the opposite mating type. For orientation relative to the neck, we measured the angle between the line from the center of the cell being scored to the centroid of the initial cluster and a line from the cell center to the center of the previous division site. Angles were grouped into segments of 30˚ increments.

### Analysis of α-factor synthesis through the cell cycle

The P$_{MFα1}$–sfGFP reporter drives synthesis of sfGFP from the MFα1 promoter. MFα1 is the major α-factor encoding gene. Average fluorescence intensity of the probe was measured from maximum projection images within an elliptical ROI drawn around each cell. Intensity values

were normalized to the value at the end of G1 by dividing by the intensity at the time of bud emergence (for cells with >1 cell cycle, the first bud emergence was used).

Because sfGFP is a stable protein, its levels do not accurately track synthesis rate. However, we can estimate the reporter synthesis rate variation using a model that incorporates the known kinetics of fluorophore maturation and dilution due to volume growth of the cells. Assuming that cells in G1 (approximately 1/3 of the cell cycle) have a synthesis rate higher than that of cells in S/G2/M (approximately 2/3 of the cell cycle), we can then extrapolate what fluorescence timecourse would be observed for a given fold-change in synthesis rate by simulating the following system:

$$\frac{dP}{dt} = k_s(t) - k_m P - k_d P,$$

$$\frac{dR}{dt} = k_m P - k_d R,$$

$$k_s(t) = \begin{cases} F, & \text{G1 stage of cell cycle} \\ 1, & \text{S/G2/M stage of cell cycle} \end{cases}$$

where $F$ = fold-change in synthesis rate; $k_m$ = fluorophore maturation rate constant (estimated at 0.123 $\text{min}^{-1}$ for sfGFP [70]); and $k_d$ = dilution rate constant (estimated at 0.007 $\text{min}^{-1}$ for a 100 min doubling time). These parameters govern the dynamics of the reporter $R$ that becomes fluorescent upon maturation of the nonfluorescent protein $P$.

## Analysis of MAPK activity

MAPK activity was measured using maximum projection fluorescent images of the sensor Ste7–NLS–NLS–mCherry. As demonstrated in [46], the sensor relocates from the nucleus to the cytoplasm upon phosphorylation by Fus3 or Kss1, and the cytoplasmic-to-nuclear ratio of the sensor reflects the MAPK activity. We used the CV of pixel intensities measured from maximum projection images to approximate the nuclear-to-cytoplasmic ratio of the probe. The CV was quite variable from cell to cell, but that variability could be limited by normalization. To approximate MAPK activity (m), (1) an elliptical ROI was drawn around each cell at each time point using ROI_TOI_QUANT_V8. (2) CV was measured for each cell for the 60 min prior to fusion. (3) CV was normalized to 1 at the time point before fusion by dividing all values by the CV at that time point. (4) Because CV falls as MAPK activity rises, activity was scored as

$$m_t = 2 - \frac{CV_t}{CV_{time\ of\ fusion}}.$$

## Analysis of receptor distribution

Membrane distribution of Ste2–sfGFP and Bem1–tdTomato were measured from medial plane fluorescent images. Using FIJI software, fluorescence intensity was averaged across the width of a 3-pixel–wide line tracing the membrane of each cell, drawn with the freehand tool. For comparisons of peak location (Fig 6G), the values for individual linescans were normalized by subtracting the background fluorescence, dividing by the maximum point in the linescan, and multiplying by 100 get the %-maximum value. For comparisons of receptor distribution (Figs 5D and 7B), the values of individual linescans were normalized by subtracting the background and bringing each trace to an integral of 1. To generate average distributions, splines

were fit to each Ste2 linescan using the smooth.spline function in R with a 0.75 smoothing factor. The normalized curves for Ste2 or Bem1 from the previous step were then centered on the maximum from the Ste2 spline fit and averaged.

### Halo assays of pheromone sensitivity

Cells were grown to mid-log phase ($OD_{600} \approx 0.4$) at 30˚C overnight in 1% yeast extract, 2% peptone, 2% dextrose (YEPD). Cultures were diluted to $2.5 \times 10^5$ cells/mL, and $5 \times 10^4$ cells were spread on YEPD plates in triplicate using sterile glass beads. Plates were allowed to dry for several minutes. For cells expressing wild-type Bar1 (Fig 6D), 2 μL of 1 mM, 500 μM, and 100 μM α-factor was spotted in three separate spots on each plate. For bar1Δ cells (Fig 7D and 7E), 2 μL of 30 μM α-factor was spotted in four spots on each plate. Plates were incubated for 48 h at 30˚C, and then images were taken using a Bio-Rad Gel Doc XR+ system (Bio-Rad, Hercules, CA, USA). Using FIJI software, circles were fit to the zone of arrest surrounding each α-factor spot, and the diameter of the circles was measured in pixels.

### Immunoblotting

Cell cultures were grown in triplicate overnight to mid-log phase in YEPD. $10^7$ cells were collected by centrifugation, and protein was extracted by TCA precipitation as described [75]. Electrophoresis and western blotting were performed as described [76]. Polyclonal anti-Cdc11 antibodies (sc-7170; Santa Cruz Biotechnology, Dallas, TX, USA) were used at 1:5,000 dilution and monoclonal mouse anti-GFP antibodies (11814460001; Roche, Basel, Switzerland) were used at 1:2,000 dilution. Fluorophore-conjugated secondary antibodies against mouse (IRDye 800CW goat anti-mouse IgG, 926–32210; LI-COR, Lincoln, NE, USA) and rabbit (Alexa Fluor 680 goat anti-rabbit IgG, A21076; Invitrogen, Carlsbad, CA, USA) antibodies were used at 1:10,000 dilution. Blots were visualized and quantified with the ODYSSEY imaging system (LI-COR). All values were normalized to a Cdc11 loading control.

### Particle-based simulations of the ratiometric and nonratiometric gradient sensing

Simulations of the ratiometric and nonratiometric models were performed using the Smoldyn software (v2.56) on Mac (3.4 GHz Intel processor) and Linux systems (2.50 GHz and 2.30 GHz Intel processors, Longleaf cluster at UNC Chapel Hill, Chapel Hill, NC, USA) [77,78]. The main components of the code are publicly available at https://github.com/mikepab/ratiometric-gpcr-particle-sims. Unless otherwise noted, the simulations were performed using the following conditions: 1) 10,000 receptor molecules and 2,500 G proteins diffusing as point particles on a sphere with diameter 5 μm; 2) the G-protein diffusion coefficient was $D = 0.002$ μm$^2$/s, and receptors were not allowed to diffuse (but see the section Particle-based simulations of receptor gradient degradation); and 3) for second-order reactions, the lambda-rho algorithm, with a fixed reactive radius ($\rho = 4$ nm) and fixed reaction probability ($P_\lambda = 1$ per simulation step), was used to compute rate constants. A reaction probability of 1 results in diffusion-limited reactions. We also studied reaction-limited versions of our models and found similar results S3B Fig). The simulation time step was set to 100 ms so that the root mean-squared displacements were below the reactive radius. The ratiometric models had a bimolecular G-protein inactivation reaction dependent upon inactive (pheromone-free) receptor, while the nonratiometric models had a unimolecular G-protein inactivation reaction that occurred with a single rate constant throughout the cell.

## Establishing receptor density and activity gradients

Receptor density and activity gradients were established prior to performing simulations using inverse transform sampling. A desired gradient (receptor or activity) was used to produce a probability distribution as a function of the spatial coordinates. A random number $P_i$ approximately Unif(0,1) was drawn for each receptor with proposed coordinates $(x_i, y_i, z_i)$, and if $P_i < P(x_i, y_i, z_i)$, a receptor was placed at the specified location (for density gradients) or was activated (for activity gradients).

## Calibrating G-protein inactivation rates

We determined inactivation rates for the nonratiometric model that produced active G protein equivalent to the inactivation rates specified for the ratiometric model (S3A Fig). Simulation-based calibration was used to determine these first-order rates rather than analytic equations for relating microscopic reaction probabilities and macroscopic rates because such equations to relate the two quantities can break down on membranes in the diffusion limit [79,80]. For consistency, the same calibration process was done for the reaction-limited versions of our simulations.

## Particle-based simulations of receptor gradient degradation

Neither the ratiometric nor nonratiometric simulations exhibited noticeable loss in gradient sensing capability when the receptor diffusion was increased from D = 0 to D = 0.0005 μm$^2$/s (S4A Fig), leading us to question whether the receptor gradient was actually degraded by diffusion over the 10-min timescale of interest. To test this, we removed the G proteins from the simulations to reduce computational costs and varied the receptor diffusivity in extended simulations (2,000 seconds, or >30 min) with a 40% to 60% receptor activity gradient and no density gradient. The active receptor gradient was measured by linear regression of the number of molecules detected in 250 nm bins along the direction of the initial gradient (S2B Fig).

## Statistical analysis

$t$ Tests were performed in Microsoft Excel via the "t-Test: Two-Sample Assuming Unequal Variances" function (Figs 6D and 7C–7E). Two-sample Kolmogorov–Smirnov tests were performed using the Real Statistics Resource Pack software (Release 5.4, developed by Charles Zaiontz) Add-in for Microsoft Excel (Figs 2C, 4A–4C, 6E, 6F, 7F, 7G, 8C and 8D). $p$-Values over 0.05 were reported as "not significant," and $p$-values under 0.05 were reported as "$p < 0.05$." $n$-values represent biological replicates except where otherwise specified. For all live-cell microscopy experiments, $n$-values represent the number of individual cells analyzed.

## Supporting information

**S1 Fig. MAPK sensor activity metric.** Cells harboring Ste7$_{1-33}$–NLS–NLS–mCherry were imaged for 150 min with 2-min resolution. (**A**) CV of Ste7$_{1-33}$–NLS–NLS–mCherry, measured from maximum projection images in an ROI encompassing the full cell. Time was normalized to "% cell cycle," with the first cytokinesis for each cell aligned at 0, and the second cytokinesis aligned at 100. (**B**) Maximum (blue) and minimum (orange) CV versus mean fluorescence intensity for each cell in (A). Mean fluorescence intensity was measured in the same ROI as the CV and averaged across all time points for each cell. (**C**) Ste7$_{1-33}$–NLS–NLS–mCherry CV measured as in (A) for mating cells. For each cell, fusion was designated as 0 min, and the timeline extends back 60 min. (**D**) MAPK activity metric plotted for the same cells shown in (C). For each cell, the CV at time point before fusion was normalized to 1. Normalized CV

values were then subtracted from 2 to generate a MAPK activity metric that rises to a value of 1 just before fusion. Strains: DLY22259 (A–D). CV, coefficient of variation; MAPK, mitogen-activated protein kinase; NLS, nuclear localization sequence; ROI, region of interest; Ste, sterile.
(TIF)

**S2 Fig. Effects of receptor diffusion in particle simulations.** (**A**) Snapshots of the active receptor gradient at $t = 0$ (black) and 2,000 s (red) for different values of the diffusion coefficient. Each curve represents a histogram with 250 nm bins derived from a single simulation. (**B**) Decay of the active receptor gradient as measured by the slopes of linear regressions fitted to the data in (A). The results show the mean of 10 realizations ± 1 SD for the four diffusion coefficients tested. Code and key data are available at https://github.com/mikepab/ratiometric-gpcr-particle-sims.
(TIF)

**S3 Fig. Calibration of G-protein inactivation rates for model comparison, and effect of diffusion-limited versus reaction-limited regimes.** (**A**) G-protein inactivation rate constant calibration, relating the nonratiometric and ratiometric models. The results shown are for the mean of 10 simulations for each condition, and the error bars represent ± 1 SD. Changing the number of receptor molecules (N) requires recalibration of the inactivation rate in the nonratiometric model. (**B**) Effect of decreasing the reaction rates to a reaction-limited regime ($P_\lambda = 0.0001$ per time step). The corresponding nonratiometric G-protein inactivation rate was $k = 0.0031$ s$^{-1}$. The results shown are for 50 realizations of each model. Although it now takes longer for simulations to reach steady state, once at steady state, the G-protein distributions are similar to those in the diffusion-limited scenario. Code and key data are available at https://github.com/mikepab/ratiometric-gpcr-particle-sims.
(TIF)

**S4 Fig. Robustness of simulation results to varying receptor abundance and diffusion.** (**A**) Accuracy of G-protein activity gradients for the nonratiometric (blue) and ratiometric (orange) models with uniform receptor density, as in Fig 8E but allowing receptor diffusion at $D = 0.0005$ μm$^2$/s. Left: illustrative simulation with measurements every 10 seconds. Right: Variability in orientation angle from 10 simulations of each model, at $t = 100$ s and 600 s snapshots (SD). (**B**) Effect of decreasing receptor abundance. Variability in orientation angle from 50 simulations of each condition. Code and key data are available at https://github.com/mikepab/ratiometric-gpcr-particle-sims.
(TIF)

**S5 Fig. Ste2–sfGFP abundance.** Uncropped western blot used to generate Fig 7C. α-GFP antibodies (green) label two bands—full-length Ste2–sfGFP and vacuolar sfGFP (note absence of vacuole signal for Ste2$^{7XR-GPAAD}$). α-Cdc11 antibodies (red) label Cdc11 (loading control). Cdc, cell division control; GFP, green fluorescent protein; sf, superfolder; Ste, sterile.
(TIF)

**S1 Video. Bem1 polarization in a mating mixture.** Cells harboring Bem1-GFP (MATα) and Bem1-tdTomato (MAT**a**) were mixed on an agarose slab and immediately imaged. (Left) False color movie of maximum projection fluorescent images of Bem1-GFP (green) and Bem1-tdTomato (magenta) in a typical mating mixture. (Right) The same movie in inverted grayscale, with labels for budding cells (red dots), G1 phase α cells (green dots), G1 phase **a** cells (teal dots), and zygotes (circled in blue). 118 min with 2-min interval between frames. Strains: DLY12943, DLY7593. Bem1, bud emergence 1; GFP, green fluorescent protein; MAT, mating

type; tdTomato, tandem dimer tomato.
(AVI)

**S2 Video. Bem1 and Spa2 polarization in mating cells.** MAT**a** cells harboring both Bem1-GFP and Spa2-mCherry were mixed with wild-type MATα cells and immediately imaged. (Top) Maximum projection fluorescent images of Bem1-GFP polarization in three example cells from cytokinesis (frame 1) through fusion with a mating partner. (Bottom) Spa2-mCherry polarization in the same three cells. 100 min with 2-min interval between frames. Strains: DLY21379. Bem1, bud emergence 1; GFP, green fluorescent protein; MAT, mating type; Spa2, spindle pole antigen 2.
(AVI)

**S3 Video. MAPK sensor in cycling cells.** The nuclear-to-cytoplasmic ratio of the MAPK sensor fluctuates through the cell cycle, rising during cytokinesis, and falling during bud growth. Fluorescent images of a field of cells harboring $Ste7_{1-33}$–NLS–NLS–mCherry. 150 min with 2-min interval between frames. Strains: DLY22259. MAPK, mitogen-activated protein kinase; NLS, nuclear localization sequence; Ste, sterile.
(AVI)

**S4 Video. MAPK sensor in mating cells.** MAPK activity rises (i.e., nuclear-to-cytoplasmic ratio of the MAPK sensor falls) as cells prepare to mate. Fluorescent images of three mating type **a** cells harboring $Ste7_{1-33}$–NLS–NLS–mCherry, mating with wild-type mating type α cells. 80 min with 2-min interval between frames. Fusion occurs in the final frame for all mating pairs. Strains: DLY22259. MAPK, mitogen-activated protein kinase; NLS, nuclear localization sequence; Ste, sterile.
(AVI)

**S1 Text. Simple mathematical model of G-protein activity.**
(PDF)

**S1 Data. All data used to generate figs.**
(XLSX)

## Acknowledgments

We thank Serge Pelet (UNIL, Switzerland), Alejandro Colman-Lerner (University of Buenos Aires, Argentina), and Patrick Ferree (Duke University) for providing plasmid reagents. Thanks to Stefano Di Talia and Amy Gladfelter, as well as members of the Lew lab, for stimulating conversations and comments on the manuscript.

## Author Contributions

**Conceptualization:** Nicholas T. Henderson, Michael Pablo, Debraj Ghose, Manuella R. Clark-Cotton, Timothy C. Elston, Daniel J. Lew.

**Data curation:** Nicholas T. Henderson.

**Formal analysis:** Nicholas T. Henderson, James Nolen.

**Funding acquisition:** Timothy C. Elston, Daniel J. Lew.

**Investigation:** Nicholas T. Henderson, Michael Pablo, Debraj Ghose, Manuella R. Clark-Cotton.

**Methodology:** Nicholas T. Henderson, Debraj Ghose, Trevin R. Zyla, Timothy C. Elston.

**Project administration:** Daniel J. Lew.

**Resources:** Trevin R. Zyla, Daniel J. Lew.

**Software:** Michael Pablo.

**Supervision:** Timothy C. Elston, Daniel J. Lew.

**Writing – original draft:** Nicholas T. Henderson.

**Writing – review & editing:** Nicholas T. Henderson, Michael Pablo, Debraj Ghose, Manuella R. Clark-Cotton, James Nolen, Timothy C. Elston, Daniel J. Lew.

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
