## [Editor Report · Decision Letter 0]

25 Jun 2019

Dear Dr Lew, 

Thank you for submitting your manuscript entitled "Ratiometric GPCR signaling enables directional sensing in yeast" for consideration as a Research Article by PLOS Biology.

Your manuscript has now been evaluated by the PLOS Biology editorial staff [as well as by an academic editor with relevant expertise] and I am writing to let you know that we would like to send your submission out for external peer review.

**Important**: Please also see below for further information regarding completing the MDAR reporting checklist. The checklist can be accessed here: https://plos.io/MDARChecklist

Please re-submit your manuscript and the checklist, within two working days, i.e. by Jun 27 2019 11:59PM.

Kind regards,

Emma Ganley, Ph.D.,

Chief Editor

PLOS Biology

INFORMATION REGARDING THE REPORTING CHECKLIST:

PLOS Biology is pleased to support the "minimum reporting standards in the life sciences" initiative (https://osf.io/preprints/metaarxiv/9sm4x/). This effort brings together a number of leading journals and reproducibility experts to develop minimum expectations for reporting information about Materials (including data and code), Design, Analysis and Reporting (MDAR) in published papers. We believe broad alignment on these standards will be to the benefit of authors, reviewers, journals and the wider research community and will help drive better practise in publishing reproducible research. 

We are therefore participating in a community pilot involving a small number of life science journals to test the MDAR checklist. The checklist is intended to help authors, reviewers and editors adopt and implement the minimum reporting framework. 

IMPORTANT: We have chosen your manuscript to participate in this trial. The relevant documents can be located here:

MDAR reporting checklist (to be filled in by you): https://plos.io/MDARChecklist

**We strongly encourage you to complete the MDAR reporting checklist and return it to us with your full submission, as described above. We would also be very grateful if you could complete this author survey:

https://forms.gle/seEgCrDtM6GLKFGQA

Additional background information:

Interpreting the MDAR Framework: https://plos.io/MDARFramework

Please note that your completed checklist and survey will be shared with the minimum reporting standards working group. However, the working group will not be provided with access to the manuscript or any other confidential information including author identities, manuscript titles or abstracts. Feedback from this process will be used to consider next steps, which might include revisions to the content of the checklist. Data and materials from this initial trial will be publicly shared in September 2019. Data will only be provided in aggregate form and will not be parsed by individual article or by journal, so as to respect the confidentiality of responses. 

Please treat the checklist and elaboration as confidential as public release is planned for September 2019.

We would be grateful for any feedback you may have.

---

## [Decision Letter · Decision Letter 1]

31 Jul 2019

Dear Dr Lew,

Thank you very much for submitting your manuscript "Ratiometric GPCR signaling enables directional sensing in yeast" for consideration as a Research Article at PLOS Biology. Your manuscript has been evaluated by the PLOS Biology editors, an Academic Editor with relevant expertise, and by four independent reviewers (one of the reviewers chose to reveal their identity and his name appears with his review below).

As you will see the reviewers are quite positive and in light of these reviews (below), we are pleased to offer you the opportunity to address the comments from the reviewers in a revised version that we anticipate should not take you very long (most comments can be easily addressed with text edits but some experiments may be needed in response to reviewers 3 and 4). We will then assess your revised manuscript and your response to the reviewers' comments and we may consult the reviewers again.

Your revisions should address the specific points made by each reviewer. Please submit a file detailing your responses to the editorial requests and a point-by-point response to all of the reviewers' comments that indicates the changes you have made to the manuscript. In addition to a clean copy of the manuscript, please upload a 'track-changes' version of your manuscript that specifies the edits made. This should be uploaded as a "Related" file type. You should also cite any additional relevant literature that has been published since the original submission and mention any additional citations in your response. 

Before you revise your manuscript, please review the following PLOS policy and formatting requirements checklist PDF: http://journals.plos.org/plosbiology/s/file?id=9411/plos-biology-formatting-checklist.pdf. It is helpful if you format your revision according to our requirements - should your paper subsequently be accepted, this will save time at the acceptance stage.

Please note that as a condition of publication PLOS' data policy (http://journals.plos.org/plosbiology/s/data-availability) requires that you make available all data used to draw the conclusions arrived at in your manuscript. If you have not already done so, you must include any data used in your manuscript either in appropriate repositories, within the body of the manuscript, or as supporting information (N.B. this includes any numerical values that were used to generate graphs, histograms etc.). For an example see here: http://www.plosbiology.org/article/info%3Adoi%2F10.1371%2Fjournal.pbio.1001908#s5.

Upon resubmission, the editors assess your revision and assuming the editors and Academic Editor feel that the revised manuscript remains appropriate for the journal, we may send the manuscript for re-review. We aim to consult the same Academic Editor and reviewers for revised manuscripts but may consult others if needed.

We expect to receive your revised manuscript within one month. Please email us (plosbiology@plos.org) to discuss this if you have any questions or concerns, or would like to request an extension. At this stage, your manuscript remains formally under active consideration at our journal; please notify us by email if you do not wish to submit a revision and instead wish to pursue publication elsewhere, so that we may end consideration of the manuscript at PLOS Biology.

When you are ready to submit a revised version of your manuscript, please go to https://www.editorialmanager.com/pbiology/ and log in as an Author. Click the link labelled 'Submissions Needing Revision' where you will find your submission record. 

Sincerely,

Emma Ganley, Ph.D., 

Chief Editor

PLOS Biology

Reviewer remarks:

---------------

Reviewer #1 (Jeffrey Segall)

---------------

Bem1 localization was followed from birth to fusion during mating. Indecisive and committed phases were identified, with increased intensity and stabilization of Bem1 or spa2 localization during the committed phases. The indecisive exploratory phase was longer for firstborn cells while the committed phase was similar for first and second born, suggesting that stable polarization occurs only to partners in G1. Secretion of alpha factor increases during G1, consistent with stable polarization being driven by increased synthesis of mating factor during G1. Using a reporter, MAPK activity was found to increase in parallel with clustering of spa2, and spa2 clustering correlated with MAPK activity no lag within the resolution of the imaging interval. Direct activation of MAPK through expression of a membrane targeted STE5 construct led to increased polarization, consistent with high MAPK activity driving formation of the committed phase. Second born partners had higher polarized initial clusters with strong orientation towards the partner, and no orientation towards the previous cytokinesis site. Using FRAP to estimate motility, the Ste2 receptor was found to have a lower diffusion coefficient than single pass transmembrane proteins. The Ste2 receptor was asymmetrically distributed in G1 cells with up to a 3 fold difference in density around the cell. Modeling a response to a gradient indicated that the asymmetric receptor density may perturb accurate gradient sensing. Adding in the effect of sst2 to generate ratiometric sensing enabled accurate gradient sensing to occur. This was tested by replacing sst2 with a human paralogue which could not interact with Ste2, and the result was altered initial orientation of Bem1. Altering Ste2 to a more uniform distribution led to more accurate gradient sensing with the human paralogue. Ratiometric sensing reduced the indecision time for second born a cells next to an alpha partner.

In summary, this paper provides a valuable detailed evaluation of the responses of cells in normal mating conditions, and gains insight into the importance of ratiometric sensing for providing accurate polarization. One aspect that it would be interesting for the authors to provide more details on from their current data (unless I missed it) is whether there are differences in Bem1/spa2 cluster kinetics of a vs alpha cells. Also, for the STE5 construct, if the authors did made the measurement - did the construct colocalize with the spa2/Bem1 probes?

Editorial point: 

Reference on line 467 should be 37, not Bush et al 2016.

---------------

Reviewer #2

---------------

In this study the authors investigate the polarization of yeast cells in response to natural gradients of pheromone emanating from mating partners. They describe a series of elegant and instructive experiments, using an admirable combination of high quality time-lapse imaging and mathematical simulation. They report that cells initially show an exploratory “indecisive” phase or polarization, of variable timespan, followed by a more uniform committed phase that immediate precedes fusion. In addition, many cells show an initial directional bias, indicating the ability to detect and follow pheromone gradients. The authors clearly explain why this process is made difficult by the small size of yeast cells, and how it is further confounded by the non-uniform distribution of receptors. They then demonstrate how the recently-described model of ratiometric sensing by the GPCR can alleviate these intrinsic difficulties and enhance the accuracy of gradient sensing, which is then thoroughly explored and supported by further experimental and simulation approaches.

Overall, this is a lovely and insightful study. The experiments are comprehensive, and the back-and-forth exchange of findings from experiments and simulations is highly compelling. Throughout the paper, the logic and presentation are exceptionally clear and convincing. I also found it refreshing how the manuscript provides a nuanced balance between distinct mechanisms that are not mutually exclusive (i.e., exploratory polarization and spatial sensing). Ultimately, I find almost nothing to criticize, as the work has so thoroughly addressed the key points from multiple angles. I have only a few specific thoughts about the issues raised that the authors might consider discussing or incorporating into this already excellent manuscript.

Specific (minor) points.

1. Regarding the “Timing of commitment” section (Fig 2 and lines 163-187): Although the “simplest hypothesis” described is reasonable, there is a good chance it is considerably more complex and interesting, in a way that might be worth discussing. In particular, the direction of pheromone secretion might be as relevant as the pheromone expression level. That is, for the first-born cell to reach the committed phase, it might require the second-born cell to not only reach G1 but also to start tentatively showing directional pheromone secretion. This relates to the old “courtship” idea of Jackson and Hartwell, in which a “conversation” between the two mating partners allows each to know that the other is showing a reciprocal commitment, so that cells in a mixture can “pair up” efficiently. How this works is a fascinating and unsolved problem (which the current study might be on the cusp of addressing); it might involve detection by each cell of an increase in pheromone concentration in the direction in which it is polarizing. 

 To give a concrete example, I wonder what would happen if the second-born cell was unable to follow a gradient, such as a cdc24-m mutant. Would the first-born cell still be equally likely to reach the committed phase once its mutant partner reached G1, or would it do so only infrequently (i.e., when it happened to lie in the default polarization direction of its partner), because it requires a further indication that the partner is reciprocating? Similarly, in the existing data (i.e., mating mixtures filmed by the authors), there should be instances in which there is more than one potential partner for a given cell, and so it should be possible to examine the polarization behavior of the potential partner that is *not* chosen. Does it ever reach the committed phase? I suspect not. In the current Video 1, there seems one such example slightly below and right of center, and another at the very bottom-right of the field.

2. Regarding the MAPK activity sensor: 

 (a) The authors describe a variation in the basal distribution with cell cycle position (Fig 3A), but do not provide any explanation. It seems likely that the sensor is being phosphorylated by CDK activity, which is not surprising given the sequences of the phosphorylation sites and the fact that the NLS is derived from a CDK-regulated target (Swi6). It seems worth stating this directly rather than leaving it seeming mysterious.

 (b) As the authors acknowledge, it is not clear if the plateau in the MAPK sensor reflects a real plateau in MAPK activity or simply saturation of the sensor. In principle this could be resolved by comparing in parallel the dose response profiles for the sensor and for conventional MAPK phosphorylation via a western blot (e.g., to see if their EC50s are similar, or if the sensor plateaus before MAPK phosphorylation does). It is unfortunate that this was never done in the original sensor paper. It is not critical here, but if the authors wanted to draw a stronger conclusion about the observed plateau then this approach could provide a straightforward way to resolve the ambiguity.

3. Lines 350-352 describes the elevated protein levels for the Ste2-7XR-GPAAD mutant and ends with a statement that these cells were “more sensitive to pheromone”. It wasn’t clear to me what point is being made here. As written, it seems to imply that the increased sensitivity is an obvious consequence of the increased protein level. But this is contrary to the ratiometric model, which explains why sensitivity does NOT increase in proportion to receptor number. Presumably the very slight increase in sensitivity observed does not match that predicted for a 4-5x increase in receptor (which, without the ratiometric mechanism, should cause cells to behave as though they are detecting 4-5x more pheromone). And presumably the increase is much less than would be observed in the hsRGS4 background, where the ratiometric mechanism is absent. So, maybe the authors should clarify if they think the “more sensitive” response violates the ratiometric model or is instead a mild-but-measurable level that perhaps indicates that the ratiometric mechanism is not perfect (or not able to function at all stimulus levels).

4. Lines 488-498 discusses some comparisons to other systems. It might be interesting to compare/contrast the results of ratiometric sensing with the “LEGI” (Local Excitation Global Inhibition) models prevalent in other chemotaxis models. The ratiometric mechanism provides an interesting counterpart because the inhibition is not “global” but instead is localized in a pattern that is reciprocal to the excitation (perhaps “LERI”, for Local Excitation Reciprocal Inhibition?). This might be an even more effective mechanism of gradient amplification than LEGI.

5. Lines 499-510: I am a bit skeptical of this suggested rationale for the role of receptor endocytosis. First, pheromone exposure stimulates the rate of receptor endocytosis. This seems the opposite of the expectation from the author’s proposal – i.e., if the main point was to clear receptors prior to mating type switching, it would make sense to rapidly clear receptors when cells were not responding to pheromone, and stop clearing them when cells were arrested in G1 and in the process of mating. Second, endocytosis also helps polarize the receptors as cells build the mating projection and grow toward the partner. This seems likely to hint at the real physiological utility. Perhaps receptor polarization helps keep the response directionally persistent, or helps contribute to the “committed” phase that the authors report here. The experiments here with the non-endocytosed receptor (Ste2-7KR) only addressed the initial directional bias and did not comment on whether the later “committed” phase was altered, but older studies reported that endocytosis-defective receptors caused mating projections to be broadened (Konopka 1988). It also seems possible that rapid endocytosis and polarization of receptors during the middle-later stages could help place a demand for increasingly focused secretion of pheromone from a partner cell, which could help cells determine if a partner is mutually reciprocating (see point #1). So, perhaps it is worth considering and discussing some additional, alternative physiological rationales.

---------------

Reviewer #3

---------------

In this manuscript the authors explore the mechanism of chemotropic signaling during mating in the budding yeast, Saccharomyces cerevisiae. Yeast cells respond to the pheromone concentration gradient by reorienting their growth axis towards the mating partner. The central question that is addressed concerns the accuracy of the internal representation of the pheromone gradient. Specifically, if ligand binding to the receptor results in an internal spatial cue, then the cue should be proportional only to the external concentration of the ligand, and not to differences in the concentration of the receptor. This is a particular problem for budding yeast in that the pheromone receptor is not initially distributed uniformly over the cell surface. The major contribution of this manuscript is to test the hypothesis that the cells use a “ratiometric” signaling method in which the local fraction of occupied receptor defines the internal spatial cue, rather than the total amount of occupied receptor. This manuscript builds on a previous paper which showed that the pheromone response is “robust” to large changes in concentration. Robustness was attributed to a ratiometric signaling model, based on data showing that Sst2p, which deactivates the G-protein, is recruited to unoccupied receptor. The current paper extends the previous one, by showing that the ratiometric model addresses the problem of non-uniform receptor distribution, provided that the receptor complexes are sufficiently clustered, such that the unoccupied receptor will lead to inactivation of nearby activated receptor. The overall effect would make internal polarization more aligned with the external gradient and sharpen the internal representation of the signal. In addition, this paper examines the response in cells with biological levels of the receptors, and examine polarization throughout the course of mating.

The first part of the paper concerns characterization of the kinetics of polarization, during encounters between cells of opposite mating type. The authors discern a period of “indecision” in which the polarity markers, Bem1-GFP and Spa2-GFP, are broadly distributed and/or mis-oriented, followed by a period of “commitment” in which the markers cluster more strongly close to the mating partner. The commitment phase is correlated with high levels of MAP-kinase activity, based on a nuclear localization assay. Presumably commitment reflects the overall effect of feedback loops cementing the chosen polarization.

The remaining parts of the paper test various aspects of the model, by looking at polarization in mutants that replace Sst2 with the human homolog, which doesn’t interact with the receptor, or that express the receptor more uniformly, or in a more clustered fashion. Consistent with the model, cells in which the Ssst2 homolog are expressed are significantly defective in orienting with the gradient, whereas the amount or distribution of the receptor does not affect orientation, with one notable exception. In cells in which the receptor was uniformly distributed, Sst2 was not required for initial orientation, highlighting the problem of receptor non-uniformity.

On the whole the paper is well written and the experiments are well designed and executed. The adaption of the ratiometric model to the problem of receptor clustering is a significant advance, and shows that the model address both problems of receptor variation, spatial and concentration. The coupling of modeling with experiment is a real strength of the paper.

A few major concerns:

1. A model is presented that hypothesizes that cells synchronize their commitment to occur in G1 by increasing the expression of pheromone in G1. The experiment to establish this is to use the MFalpha1 promoter to drive expression of intracellular GFP. The authors show that there is a saw-toothed temporal pattern in GFP, decreasing by about 20-30% during S/G2/M. Although this is consistent with G1 specific expression, it is a difficult experiment to interpret, given that the volume of the cell is changing in a non-linear manner. There is no control demonstrating what steady state expression of an intracellular GFP actually looks like, so it is hard to know whether it reflects cell cycle regulated expression, and dilution, or some other effect. Moreover, the magnitude of the effect is relatively small, which decreases the confidence that this is an important effect. Most of the difficulty stems from examining accumulation of a stable protein. This point needs to supported by a more direct examination of the mRNA. I note that there is no data at SGD on the cell cycle regulation of this promoter. 

2. All of the modeling seems to be based on cells responding to stable preformed pheromone gradients. However, the experiments are typically based on cells of opposite mating type being put in close proximity at t=0, and there is no consideration in the modeling for a temporal component effecting the response as the pheromone starts to diffuse between the cells. This may not affect the modeling significantly, but this needs to be addressed.

3. It has recently been shown that there are changes in the localization of components of the polarization apparatus late in mating, just prior to cell fusion, and dependent on cell fusion proteins. It is unclear what role the cell fusion proteins play in the commitment phase. While new experiments are likely to be beyond the scope of the paper, the role of cell fusion proteins late in the pathway should be mentioned and discussed in the introduction and discussion.

---------------

Reviewer #4

---------------

In their manuscript titled ‘Ratiometric GPRC signaling enables directional sensing in yeast’ Henderson et al analyze directional pheromone sensing in a co-culture of a and alpha mating type budding yeast. Adjacent yeast of opposing mating type exhibit a variety of dynamic polarization phenomena that are accurately quantified and categorized as they orient towards one another. It was found that ratiometric sensing, whereby bound receptors produce positive and unbound receptors produce negative signals, was important for cells to able to polarize towards their prospective partner despite having a non-uniform concentration of pheromone receptors on their surface (intuitively, if sensing were not ratiometric, then there would be a bias towards where there is more receptor). Importantly, the authors are examining the properties of the pheromone response and polarization pathways in much less synthetic context than usual (which usually just controlling the extracellular pheromone concentration). Here, they are examining the response of a and alpha yeast grown together in a co-culture, which is one large step more physiological, and the authors can examine the interesting mating dynamics of a sort of call and response of two potential mating cells. This paper is truly a tour de force. The mutant analysis in Fig 6 and 7 where the negative signal for unbound receptors is broken via the use of the human version of Sst2 is awesome. 

There are so many exciting results packed into the 8 figures of this paper, and the experiments are quite carefully controlled, that the manuscript can be published as is in PLoS Biology. I have a few suggestions though that the authors may want to consider, but their clarification is not essential publication. 

In Fig. 2D, the signal for ‘pheromone production’ goes from ~0.8 to ~1, about a 20% fluctuation through the cell cycle. This is not much and probably an understatement of the reality, The authors could either used a destabilized fluorescent protein or make a model to estimate protein synthesis and dilution through the cell cycle for individual cells to get a better estimate of that the pheromone production actually is. 

In Fig 3A the authors state and show images of fluctuations of the MAPK sensor through the cell cycle. It would be nice if this were quantified and if we knew of this was due to Cdk activity (which can target similar sites) or due to fluctuations in the basal MAPK activity. Obviously, this is not a paper about the sensor, but in the case of these sensors, the more we know about specificity the better. If the fluctuation in the cell cycle is due to Cdk, that does not affect the author’s conclusions about the sensor dynamics in pheromone (where there is no Cdk activity). Would an analog sensitive Fus3 be useful here?

In Figure 6H, the authors should quantify the images as they have for other signals to show the colocalization here of Bem1 and Ste2.

---

## [Editor Report · Decision Letter 2]

6 Sep 2019

Dear Dr Lew,

Thank you for submitting your revised Research Article entitled "Ratiometric GPCR signaling enables directional sensing in yeast" for publication in PLOS Biology. The Academic Editor has now carefully evaluated your revision. We're delighted to let you know that we're now editorially satisfied with your manuscript. Please do however see below for Data Policy-related requests.

Furthermore, before we can formally accept your paper and consider it "in press", we also need to ensure that your article conforms to our guidelines. A member of our team will be in touch shortly with a set of requests. As we can't proceed until these requirements are met, your swift response will help prevent delays to publication.

Please note that you may have the opportunity to make the peer review history publicly available. The record will include editor decision letters (with reviews) and your responses to reviewer comments. If eligible, we will contact you to opt in or out.

Early Version

Sincerely,

Hashi Wijayatilake, PhD, 

Managing Editor

PLOS Biology

DATA POLICY:

You are aware of the PLOS Data Policy, which requires that all data be made available without restriction: http://journals.plos.org/plosbiology/s/data-availability. For more information, please also see this editorial: http://dx.doi.org/10.1371/journal.pbio.1001797

** Thank you for providing most of your data in the 'Raw Data Supplement' file (please do re-name this file according to the convention noted above). Please do also ensure that you provide the individual numerical values that underlie the summary data displayed in all the following figure panels (which are not currently present in your data file) as they are essential for readers to assess your analysis and to reproduce it:

Figs. 2DG, 4DE, 5E-G, 6B, 8BCDE, S2AB, S3AB, S4AB

** Please also ensure that the figure legends in your manuscript include information on where the underlying data can be found.

For manuscripts submitted on or after 1st July 2019, we require the original, uncropped and minimally adjusted images supporting all blot and gel results reported in an article's figures or Supporting Information files. We will require these files before a manuscript can be accepted so please prepare them now, if you have not already uploaded them. Please carefully read our guidelines for how to prepare and upload this data: https://journals.plos.org/plosbiology/s/figures#loc-blot-and-gel-reporting-requirements.

---

## [Editor Report · Decision Letter 3]

25 Sep 2019

Dear Dr Lew,

On behalf of my colleagues and the Academic Editor, Sophie G. Martin, I am pleased to inform you that we will be delighted to publish your Research Article in PLOS Biology. 

Early Version

PRESS 

Kind regards,

Hannah Harwood

Publication Assistant, 

PLOS Biology

on behalf of

Hashi Wijayatilake,

Managing Editor

PLOS Biology